# Automated annotation of birdsong with a neural network that segments spectrograms

**Yarden Cohen[1]\*[†], David Aaron Nicholson[2][†], Alexa Sanchioni[3][‡], Emily K Mallaber[3][‡], Viktoriya Skidanova[3][‡], Timothy J Gardner[4]\***

[1]Department of Brain Sciences, Weizmann Institute of Science, Rehovot, Israel; [2]Biology department, Emory University, Atlanta, United States; [3]Biology department, Boston University, Boston, United States; [4]Phil and Penny Knight Campus for Accelerating Scientific Impact, University of Oregon, Eugene, United States

**Abstract** Songbirds provide a powerful model system for studying sensory-motor learning. However, many analyses of birdsong require time-consuming, manual annotation of its elements, called syllables. Automated methods for annotation have been proposed, but these methods assume that audio can be cleanly segmented into syllables, or they require carefully tuning multiple statistical models. Here, we present TweetyNet: a single neural network model that learns how to segment spectrograms of birdsong into annotated syllables. We show that TweetyNet mitigates limitations of methods that rely on segmented audio. We also show that TweetyNet performs well across multiple individuals from two species of songbirds, Bengalese finches and canaries. Lastly, we demonstrate that using TweetyNet we can accurately annotate very large datasets containing multiple days of song, and that these predicted annotations replicate key findings from behavioral studies. In addition, we provide open-source software to assist other researchers, and a large dataset of annotated canary song that can serve as a benchmark. We conclude that TweetyNet makes it possible to address a wide range of new questions about birdsong.

**\*For correspondence:**
yarden.j.cohen@weizmann.ac.il (YC);
timg@uoregon.edu (TJG)

[†]These authors contributed equally to this work
[‡]These authors also contributed equally to this work

## Editor's evaluation

Animals create an enormous diversity of sounds. To study the neural basis or behavioral logic of animal communication, it is first necessary to categorize sounds into distinct types. Here, the authors create a novel neural network that includes an LSTM to enable automated annotation of massive birdsong datasets. This widely usable method will have a big impact in the birdsong field and, more generally, will provide an ascendant generation of scientists with yet another example of how machine learning methods are revolutionizing the rigorous study of animal behavior.

## Introduction

Songbirds are an excellent model system for investigating sensory-motor learning and production of sequential behavior. Birdsong is a culturally transmitted behavior learned by imitation (*Mooney, 2009*). Juveniles typically learn song from a tutor, like babies learning to talk. Their songs consist of vocal gestures executed in sequence (*Fee and Scharff, 2010*). In this and many other ways, birdsong resembles speech (*Brainard and Doupe, 2002*). A key advantage of songbirds as a model system is that birds sing spontaneously, producing hundreds of song bouts a day. Their natural behavior yields a detailed readout of how learned vocalizations are acquired during development and maintained in adulthood. Leveraging this amount of data requires methods for high-throughput automated

analyses. For example, automated methods for measuring similarity of juvenile and tutor song across development (*Tchernichovski et al., 2000*; *Mets and Brainard, 2018a*) led to important advances in understanding the behavioral and genetic bases of how vocalizations are learned (*Tchernichovski et al., 2001*; *Mets and Brainard, 2018b*; *Mets and Brainard, 2019*). However, similarly scaling up other analyses of vocal behavior is currently hindered by a lack of automated methods.

A major roadblock to scaling up many analyses is that they require researchers to annotate song. Annotation is a time-consuming process done by hand with graphical user interface (GUI) applications, for example Praat, Audacity, Chipper (*Boersma and Weenink, 2021*; *Audacity Team, 2019*; *Searfoss et al., 2020*). To annotate birdsong, researchers follow a two-step process (*Thompson et al., 2012*; *Kershenbaum et al., 2016*). First, they segment song into units, often called syllables, and second, they assign each syllable a label. Labels correspond to a set of discrete syllable classes that a researcher defines for each individual bird. Many models and analyses rely on song annotated at the syllable level, including: statistical models of syntax (*Markowitz et al., 2013*; *Jin et al., 2011*;

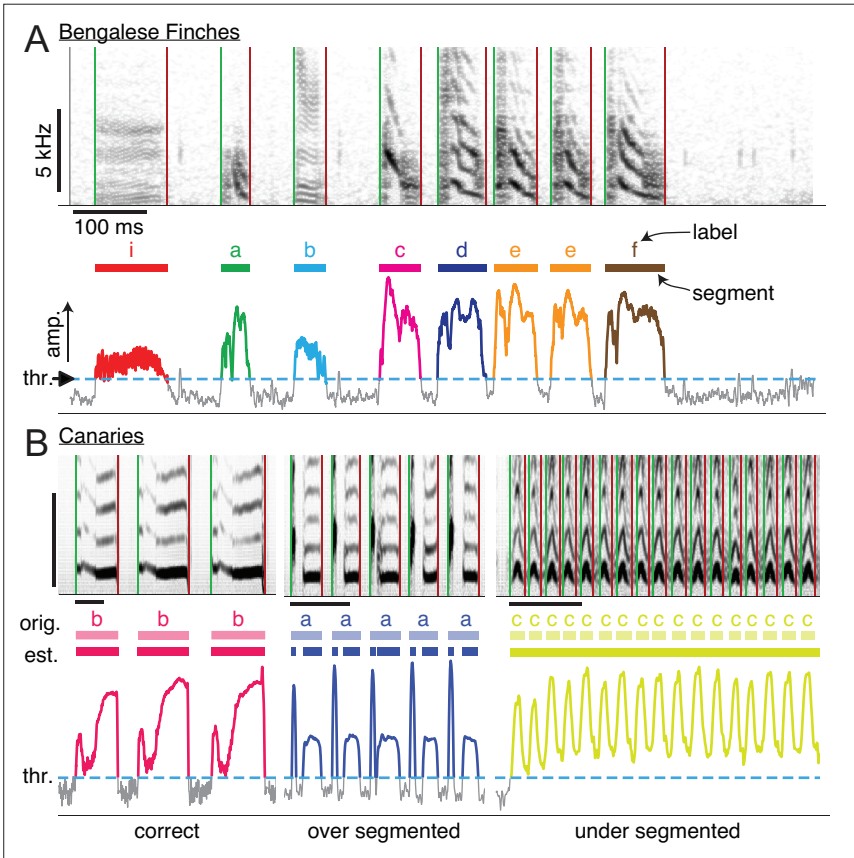

**Figure 1.** Manual annotation of birdsong. (**A**) Schematic of the standard two-step process for annotating song by hand (e.g. with a GUI application). Top axes show a spectrogram generated from a brief clip of Bengalese finch song, with different syllable types. Middle and bottom axes show the steps of annotation: first, segments are extracted from song by setting a threshold ('thr.', dashed horizontal line, bottom axes) on the amplitude and then finding continuous periods above that threshold (colored regions of amplitude trace, bottom axes). This produces segments (colored bars, middle axes) that an expert human annotator manually labels (characters above colored bars), assigning each segment to one of the syllable classes that the annotator defines for each individual bird. (**B**) Examples showing how the standard approach of segmenting with a fixed amplitude threshold does not work well for canary song. Above threshold amplitudes are plotted as thicker colored lines. For a fixed threshold (dotted line, bottom axes), syllables of type 'b' are correctly segmented, but syllables of type 'a' are incorrectly segmented into two components, and syllables of type 'c' are not segmented.

The online version of this article includes the following figure supplement(s) for figure 1:

**Figure supplement 1.** Example of two consecutive canary phrases that differ mostly in inter-syllable gaps.

**Figure supplement 2.** Comparison of descriptive statistics of birdsong syllables across species.

*Berwick et al., 2011*; *Hedley, 2016*); computational models of motor learning (*Sober and Brainard, 2009*; *Sober and Brainard, 2012*); and analyses that relate both acoustic features and sequencing of syllables to neural activity (*Leonardo and Fee, 2005*; *Sober et al., 2008*; *Wohlgemuth et al., 2010*). As these examples demonstrate, our ability to leverage songbirds as a model system would be greatly increased if we could automate song annotation.

Many previously proposed methods for automating annotation follow the same two-step process used when annotating manually. We describe the process in more detail, as illustrated in *Figure 1A*, to make it clear how limitations can arise when automating these two steps. First, audio is segmented into syllables by applying a widely-used simple algorithm. Basically, the algorithm consists of setting a threshold on amplitude and finding each uninterrupted series of time points above that threshold. After segmentation, manual annotation proceeds with a researcher assigning labels to syllables (letters ['i', 'a', 'b',…] in *Figure 1A*). We emphasize that each individual bird will have a unique song, even though songs are recognizably similar within a species, and that a researcher chooses an arbitrary set of labels for each individual's repertoire of syllables. This means that any automated method must be capable of reliably classifying these arbitrary classes across individuals and species.

Previous attempts to automate the annotation of birdsong kept the segmentation and labeling steps separate, and therefore suffered from limitations in each step. Methods such as semi-automatic clustering (*Burkett et al., 2015*; *Daou et al., 2012*), and supervised machine learning algorithms (*Troyer lab, 2012*; *Tachibana et al., 2014*; *Nicholson, 2016*), can fail when the song of a species is not reliably segmented using the standard algorithm just described. We illustrate this in *Figure 1B* with examples of song from canaries. One reason the standard algorithm does not work is that the amplitude of canary song varies so greatly that no single threshold reliably segments all syllables. Even for species where good segmenting parameters can be found, a given individual's song will often have one or two syllable classes that require an annotator to clean up its onsets and offsets by hand. Furthermore, other sounds in the environment, such as beak clicks and movement noise, are inevitably segmented as if they were syllables. Machine learning models operating on segmented audio will happily assign these segments a syllable class, resulting in false positives. Various other statistical methods can be used to remove these false positives, such as outlier detection algorithms. In combination with such methods, supervised machine learning models have been used to successfully annotate large-scale behavioral experiments (e.g. *Veit et al., 2021*). But these additional clean-up steps add complexity and require the researcher to perform further tuning and validation.

Automated annotation methods may also face limitations at the step of labeling segments. Many machine learning models make use of pre-defined, engineered features, that may not reliably discriminate different classes of syllables across individual birds or species. Likewise, features extracted from single syllables do not capture temporal dependencies, that if taken into account can improve the classification accuracy (*Anderson et al., 1996*; *Kogan and Margoliash, 1998*; *Nicholson, 2016*). (An example where temporal features are needed is shown in *Figure 1—figure supplement 1*.) This issue with models that do not leverage temporal information becomes particularly important for species whose song has more variable sequencing (see *Figure 1—figure supplement 2*), like the Bengalese finch and canary song we study here. Such issues likely account for why there is no prior work on algorithms for automated annotation of canary song at the syllable level. Canaries have provided unique insights into neuronal regeneration, seasonality, interhemispheric coordination, hormones, and behavior (*Goldman and Nottebohm, 1983*; *Nottebohm, 1981*; *Suthers et al., 2012*; *Wilbrecht and Kirn, 2004*; *Alvarez-Buylla et al., 1990*; *Gardner et al., 2005*). In spite of this, canary song with its rich syllable repertoire and complex song syntax (*Markowitz et al., 2013*; *Alonso et al., 2009*; *Appeltants et al., 2005*; *Alliende et al., 2013*) remains understudied, as does the similarly complex song of many other species.

Given the limitations faced by existing methods, we sought to develop an algorithm for automated annotation of syllables that (1) does not require cleanly segmented audio when predicting labels, (2) only requires training a single model, and (3) does not rely on hand-engineered features. To meet these criteria, we developed a deep neural network that we call TweetyNet, shown in *Figure 2*. Deep neural network models have the potential to address our criteria, because they can learn features from the training data itself, and they can be designed to map directly from spectrograms of song to predicted annotations, eliminating the need to segment audio. Below we test whether TweetyNet meets our criteria. To do so, we benchmark TweetyNet on Bengalese finch and canary song. We

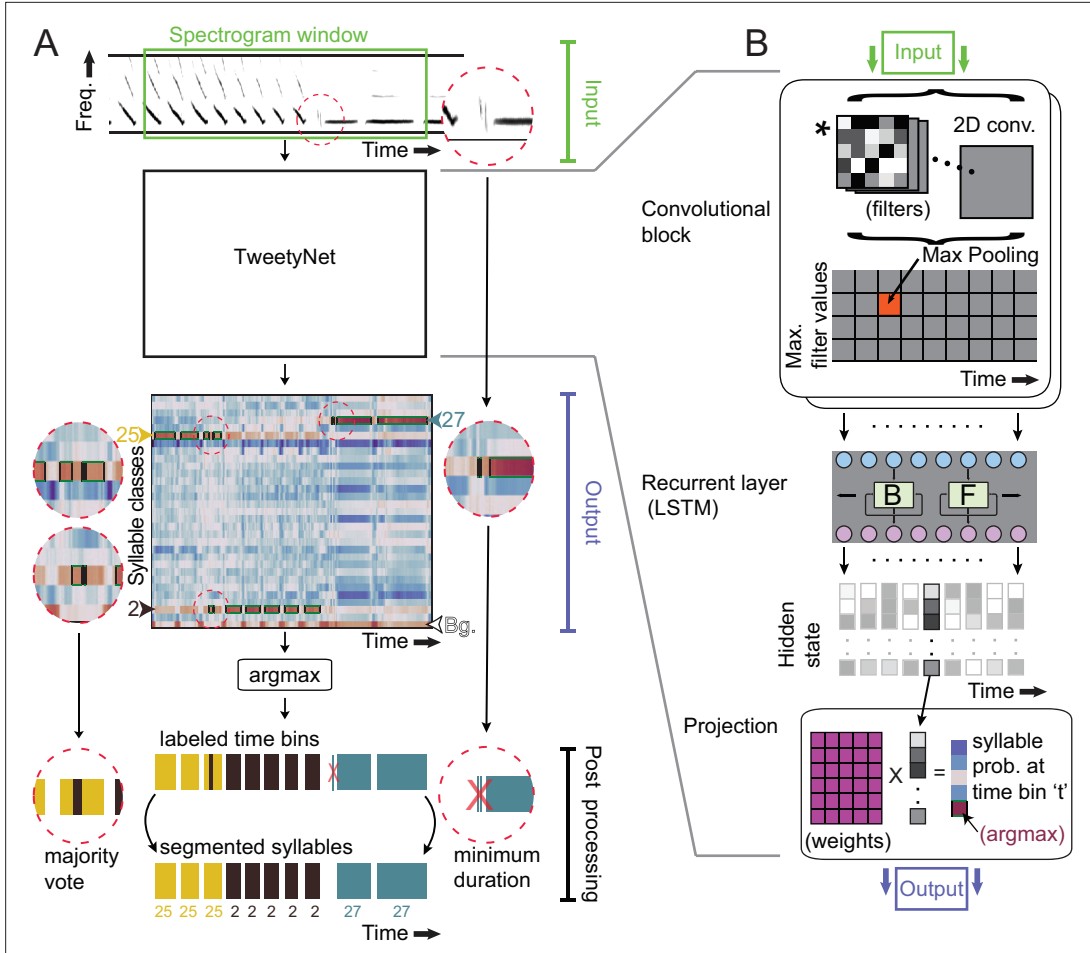

**Figure 2.** TweetyNet operation and architecture. (**A**) TweetyNet takes as input a window from a spectrogram, and produces as output an estimate of the probability that each time bin in the spectrogram window belongs to a class $c$ from the set of predefined syllable classes $C$. This output is processed to generate the labeled segments that annotations are composed of: (1) We apply the argmax operation to assign each time bin the class with the highest probability. (2) We use the 'background' class we add during training (indicated as 'Bg.') to find continuous segments of syllable class labels. (3) We post-process these segments, first discarding any segment shorter than a minimum duration (dashed circle on right side) and then taking a majority vote to assign each segment a single label (dashed circles on left side). (**B**) TweetyNet maps inputs to outputs through a series of operations: (1) The convolutional blocks produce a set of feature maps by convolving (asterisk) their input and a set of learned filters (greyscale boxes). A max-pooling operation downsamples the feature maps in the frequency dimension. (2) The recurrent layer, designed to capture temporal dependencies, is made up of Long Short Term Memory (LSTM) units. We use a bidirectional LSTM that operates on the input sequence in both the forward (F) and backward (B) directions to produce a hidden state for each time step, modulated by learned weights in the LSTM units. (3) The hidden states are projected onto the different syllable classes by a final linear transformation, resulting in a vector of class probabilities for each time bin $t$. For further details, please see section 'Neural network architecture' in Materials and methods.

demonstrate that TweetyNet achieves robust performance across species and individuals, whose song can vary widely even within a species, and across many bouts of song from one individual, i.e., across days of song. Using large datasets from actual behavioral experiments, we show that automated annotations produced by TweetyNet replicate key findings about the syntax of song in both species.

## Proposed model

First we describe our approach in enough detail to provide context. As shown in *Figure 2*, a TweetyNet model takes as input a window from a spectrogram of song, and produces as output a label for each time bin of that spectrogram window. Because it labels each time bin in a spectrogram, TweetyNet does not require segmented audio to predict annotations. In order to recover segments from the network output, we add a 'background' class to the set of syllable labels (indicated as 'Bg.' in *Figure 2A*). This class includes the brief quiet intervals between syllables, as well as noises, such as

beak clicks, wing flaps, and non-song calls. When predicting annotations for new data, we use these time bins classified as 'background' to find segmented syllables in the network's output.

## Neural network architecture

The neural network architecture we developed is most closely related to those designed for event detection, as studied with audio (*Böck and Schedl, 2012*; *Parascandolo et al., 2016*) or video (*Lea et al., 2017*) data, where the task is to map a time series to a sequence of segments belonging to different event classes. Like those previous works, TweetyNet's architecture combines two structural elements commonly found in neural networks, as shown in *Figure 2B*. The first element is a convolutional block, used in computer vision tasks to learn features from images (*Goodfellow et al., 2016*; *Farabet et al., 2013*; *Krizhevsky et al., 2012*). (The term 'block' refers to a group of operations.) The second element is a recurrent layer, often used to predict sequences (*Graves, 2012*). Specifically, we use a bidirectional Long Short-Term Memory (LSTM) layer that has been shown to successfully learn temporal correlations present in speech (*Graves, 2012*), music, and acoustic scenes (*Böck and Schedl, 2012*; *Parascandolo et al., 2016*). Importantly, we maximized the information available to the recurrent layer by choosing parameters for the pooling operation in the convolutional blocks that *did not* downsample in the temporal dimension. We made this choice based on previous work on automatic speech recognition (*Sainath et al., 2013a*; *Sainath et al., 2013b*). Please see section 'Neural network architecture' in Materials and methods for a more detailed description of the network architecture, parameters, and citations of relevant literature that motivated our design choices.

## Post-processing neural network output and converting it to annotations

In the results below, we show that we significantly reduce error by post-processing network outputs with two simple transformations. So that these results are clear, we now describe how we convert outputs to annotation, including post-processing. For each window from a spectrogram, the network outputs a matrix with shape ($c$ classes $\times$ $t$ time bins) (ignoring the batch dimension). Values along dimension $c$ are the probabilities that the network assigns to each class label. Along that dimension, we apply the $\arg\max$ operation ('argmax' in *Figure 2A*) to produce a vector of length $t$, where the value in each time bin is the class label that the network estimated had the highest probability of occurring in that time bin. We recover segments from this vector by finding all uninterrupted runs of syllable labels that are bordered by bins labeled with the 'background' class. We consider each of these continuous runs of syllable labels to be a segment.

To clean up these segments, we apply two transformations ('Post processing' in *Figure 2A*). First, we remove any segment shorter than a minimum duration, specified by a user. Second, we then take a 'majority vote' by counting how many times each label is assigned to any time bin in a segment, and then assigning the most frequently occurring label to all time bins in the segment, overriding any others. To annotate an entire spectrogram corresponding to one bout of song, we feed consecutive windows from the spectrogram into a trained network, concatenate the output vectors of labeled time-bins, and then apply the post-processing. Finally we convert the onset and offset of each predicted segment back to seconds, using the times associated with each bin in the spectrogram, and we convert the segment's integer class label back to the character label assigned by human annotators.

Using the method just described, a single TweetyNet model trained end-to-end can successfully annotate entire bouts of song at the syllable level. We are aware of only one previous study that takes a similar approach, from *Koumura, 2016*. That study evaluated pipelines combining a convolutional neural network for classifying spectrogram windows with additional models that learn to correctly predict sequences of labels (e.g. Hidden Markov Models). In contrast, TweetyNet is a single neural network trained end-to-end, meaning it does not require optimizing multiple models. That previous study also focused on annotating specific sequences of interest within a song. Here, our goal is to annotate entire song bouts, not specific sequences, so as to automate the process as much as possible.

## Results

We assess performance of TweetyNet in two ways. First, we benchmark TweetyNet as a machine learning model, adopting good practices from that literature. We use a metric that we call the syllable error rate, by analogy with the word error rate, the standard metric for automatic speech recognition.

It is an edit distance, meaning its magnitude increases with the number of edits (insertions, deletions, and substitutions) required to 'correct' the predicted sequence of labels so that it matches the ground truth sequence. (For specifics, see 'Metrics' in Materials and methods.) The edit distance is normalized, converting it into a rate, as required to measure performance across sequences of different lengths. We show syllable error rate as a percentage throughout for readability. Thus, a 1.0% syllable error rate can be thought of as 'one edit per every 100 syllable labels'. It should be noted, though, that the syllable error rate can grow larger than 100%, for example if a predicted sequence has many more labels than the original. As results below show, this metric is very informative when benchmarking a model such as ours.

The second way we study TweetyNet's performance is meant to align with the point-of-view of an experimentalist, who simply wants to know whether the annotations that TweetyNet produces are 'good enough' to answer their research question. To that end, we show that annotations predicted by trained TweetyNet models recover key findings from behavioral studies in Bengalese finches and canaries, by fitting statistical models of song syntax to predicted annotations.

## TweetyNet avoids limitations that arise from segmenting audio

To show that TweetyNet avoids issues that result from relying on segmented audio (as described in the 1 Introduction), we compare its performance with a model that predicts labels given engineered acoustic features extracted from segmented audio. Specifically, we use a Support Vector Machine (SVM) model and pre-defined features adapted from *Tachibana et al., 2014* as described in 'Comparison with a Support Vector Machine model' in Materials and methods. To compare these two

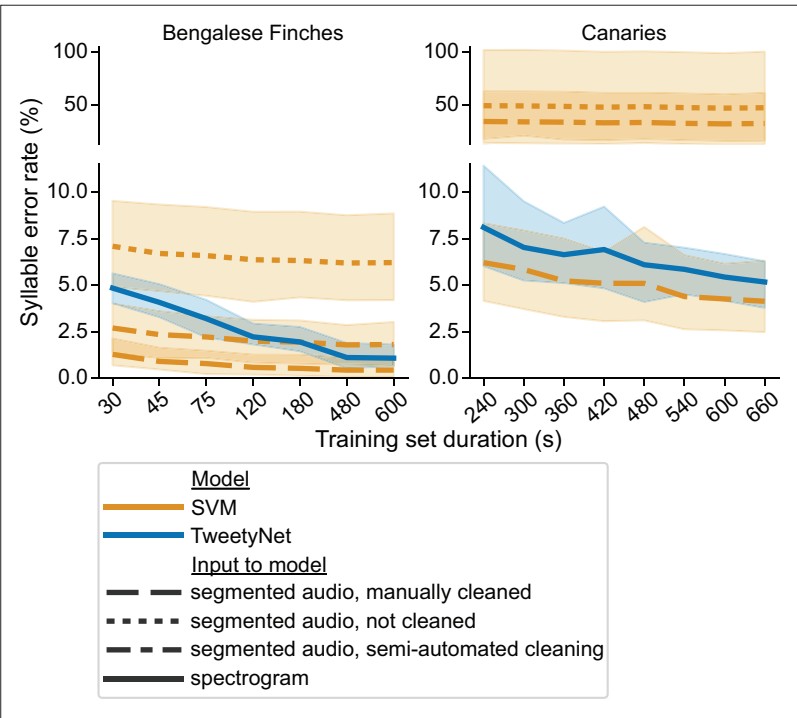

**Figure 3.** Comparison of TweetyNet with a support vector machine (SVM) model. Plots show syllable error rate (y axis) as a function of training set size (x axis, size of training set in seconds). Syllable error rate is an edit distance computed on sequences of text labels. Here it is measured on a fixed, held-out test set (never seen by the model during training). Hues correspond to model type: TweetyNet neural network (blue) or SVM (orange). Shaded areas around lines indicate the standard deviation across song of individual birds, and across model training replicates (each trained with different subsets of data randomly drawn from a total training set, n = 4 Bengalese finches, 10 replicates per bird;n = 3 canaries, 7 replicates per bird). Line style indicates input to model: spectrogram (solid line), or segmented audio, processed in three different ways, either manually cleaned by human annotators (dashed), not cleaned at all (dotted), or cleaned with a semi-automatic approach (dot-dash).

The online version of this article includes the following source data for figure 3:

**Source data 1.** Data used to generate line plots.

models we generated learning curves, that plot performance as a function of the amount of manually-annotated training data.

The core question is: how does each model perform when applied to unlabeled data that has been pre-processed as required, *without* any additional inspection or arduous manual cleaning from a human expert? For the SVM, the unlabeled data is pre-processed by segmenting the audio, while for TweetyNet, the audio files are converted to spectrograms. For both models, data is pre-processed for prediction with the exact same parameters used to pre-process training data: for example, with the same amplitude threshold used to segment audio. To simulate this for the SVM, we needed to re-segment the audio of the test set, because the segments in the ground truth annotations have been carefully cleaned by expert human annotators. We then obtained SVM predictions for these 'raw' segments. For all training set sizes, the syllable error rate of the SVM given 'raw' segments was higher than the syllable error rate of TweetyNet, as can be seen by comparing the dotted orange lines and the solid blue lines in *Figure 3*.

This estimate of syllable error rate for the SVM may seem overly pessimistic. For example, an expert human annotator could remove any non-song noises between song bouts fairly efficiently. To mimic this simple cleaning step, we removed any segments in the re-segmented audio that did not occur between the first onset and the last offset in the manually annotated, ground truth data. Our intent was to remove most of the noises that a human annotator could rapidly identify, while leaving any mis-segmented syllables that the annotator would need to carefully adjust by hand. In this setting, with semi-automated clean-up of the segments, the SVM also had a higher syllable error rate than TweetyNet across all canaries, for all training set sizes (compare dash-dotted orange line in *Figure 3* with solid blue lines). For Bengalese finches, syllable error rate of the SVM started out lower than TweetyNet, but with 10 min of training data, error for TweetyNet was lower, and this difference was statistically significant ($p < 0.001$, Wilcoxon signed-rank test). This result indicates that much of the increased syllable error rate can be attributed to imperfect segmenting of the true syllables and other noises that take place during song bouts.

We did observe that SVM models could actually achieve a very low syllable error rate, when provided with audio segments that have been manually cleaned by human annotators. SVM predictions on this perfectly clean data are lower than the syllable error rate of TweetyNet. For models trained with 10 min of data, this difference was again significant ($p < 0.05$, Wilcoxon signed-rank test). However, if applying a machine learning model required human annotators to manually clean the segments produced from audio by the standard algorithm, it would defeat the purpose of automating annotation.

Lastly, we observed that there was a much higher standard deviation in error rate, computed across individuals and training replicates, for SVM models predicting labels for uncleaned or semi-cleaned segmented audio when compared with TweetyNet ($p < 0.001$, Levene's test). The standard deviation is indicated by the shaded areas in *Figure 3*. This results shows that TweetyNet performs well across random samples of each bird's song, because each replicate was trained on a randomly drawn subset from a larger pool of training data. Our software ensured that at least one instance of each syllable class was present in those subsets (please see 'Learning curves' for details). This result suggests that experimenters will not need to carefully construct training sets of data to fit TweetyNet models, as long as they ensure that training sets contain a minimum number of instances of each syllable class.

## Tweetynet annotates with low error rates across individuals and species

The third criterion we set out above is that our model should be capable of learning the unique song of each individual. Here we show that this criterion is met by our method achieving low error across individuals and across species. To show this, we carried out further experiments, adding song from an additional four Bengalese finches from the dataset accompanying *Koumura, 2016* (see 'Annotation of Bengalese finch song' in Materials and methods for details). This gave us a dataset of song from 8 Bengalese finches recorded and annotated in two different research groups. In *Figure 4*, we show learning curves for the 8 Bengalese finches and the three canaries, this time plotting lines for each individual, to better understand how the model performs for each bird's unique song. Here, we consider the syllable error rate as defined above, and in addition the frame error, which is the fraction of time bins classified incorrectly, displayed as a percent. Results here and in the next section will

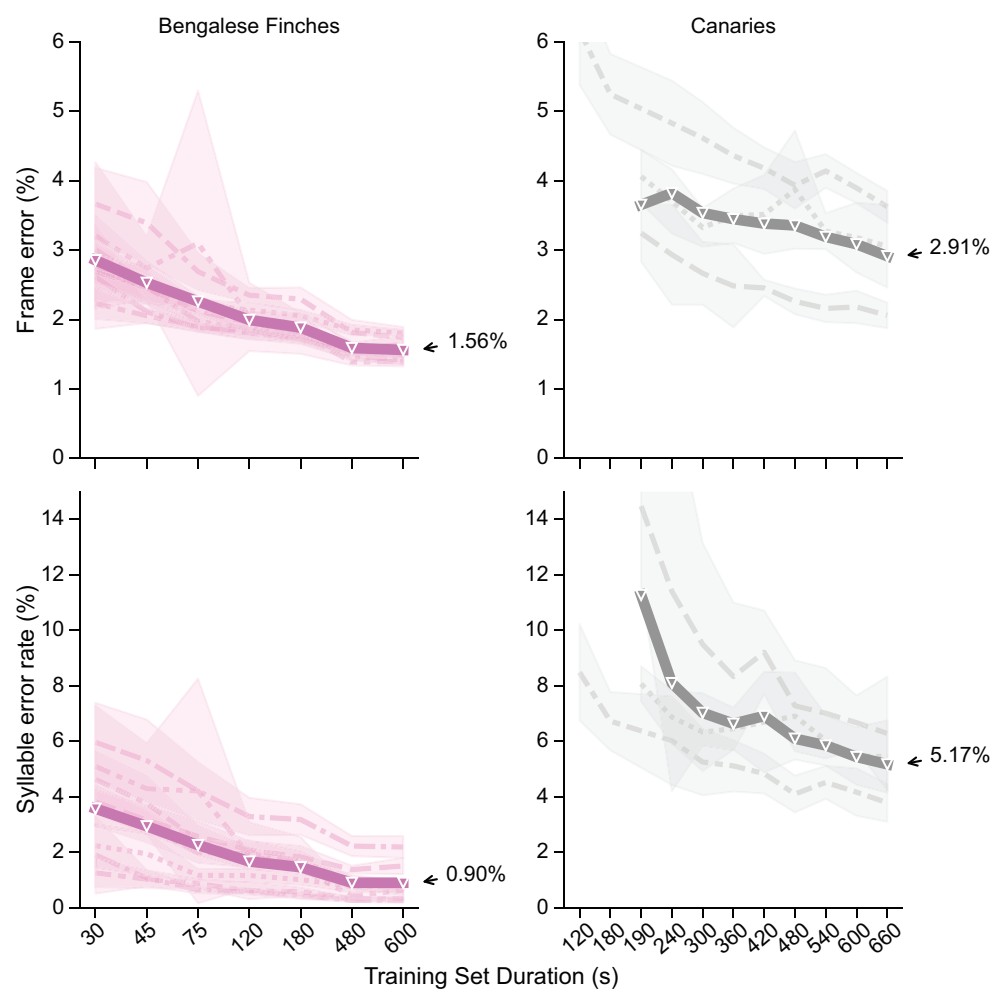

**Figure 4.** Performance of TweetyNet across songs of 8 Bengalese finches and three canaries. Plots show frame error (y axis, top row) and syllable error rate (y axis, bottom row) as a function of training set size (x axis, in seconds). Frame error is simple accuracy of labels the network predicted for each time bin in spectrograms, while syllable error rate is an edit distance computed on sequences of labels for the segments that we recover from the vectors of labeled time bins (as described in main text). Thick line is mean across all individuals, thinner lines with different styles correspond to individual birds (each having a unique song). Shaded areas around lines for each bird indicate standard deviation of metric plotted across multiple training replicates, each using a different randomly-drawn subset of the training data. Metrics are computed on a fixed test set held constant across training replicates. Here hue indicates species (as in **Figure 5A** below): Bengalese finches (magenta, left column) and canaries (dark gray, right column).

The online version of this article includes the following source data for figure 4:

**Source data 1.** Data used to generate plots for Bengalese finches.

**Source data 2.** Data used to generate plots for canaries.

demonstrate why it is important to measure both the frame error and the syllable error rate. Across all 8 Bengalese finches, the mean syllable error rate obtained by TweetyNet was 0.9%, and the mean frame error was 1.56%. It can be seen that the model performed well across most birds and training replicates, although for two birds the syllable error rate exhibited a relatively high standard deviation for training sets of size 75 s or less. Across all individuals, it appeared that 8–10 min worth of manually annotated data was the minimal amount needed to train models achieving the lowest observed syllable error rates. For canaries, with 11 min of training data, the mean syllable error rate was 5.17%, and the mean frame error was 2.91%. It was unclear from the learning curves for canaries whether the syllable error rate of TweetyNet had reached an asymptotic value at the largest training set size.

Because training models on canary song could be computationally expensive, we did not include larger data sets for these curves. To obtain an estimate of the asymptotic syllable error rate, for each bird we trained one replicate on a single dataset of 60 minutes of song (instead of training multiple replicates with randomly drawn subsets of the training data). This produced an estimated asymptotic mean syllable error rate of 3.1(± 0.2)% for TweetyNet on canary song. Taken together, these bench-marking results suggest that the syllable error rate of TweetyNet is low enough to enable automated annotation of large-scale datasets from behavioral experiments. We show this rigorously below, but first we interrogate more closely how the model achieves this performance.

## Simple post-processing greatly reduces syllable error rates

One of our criteria for an automated annotation method was that it should only require training a single model. Although our approach meets this criterion, there are of course hyperparameters for training the model that we tuned during our experiments, and there is additional post-processing applied to the model outputs when converting them to annotations. (The term 'hyperparameter' refers to parameters that configure the model, such as the batch size during training or the size of the spectrogram windows, as opposed to the parameters in the model itself, optimized by training.) Here, we take a closer look at how post-processing and hyperparameters impact performance, to understand how TweetyNet works 'under the hood', and to provide a starting point for users applying the model to their own data.

As described above, the post-processing consists of first discarding any segments shorter than a minimum duration, and then taking a 'majority vote' within any consecutive run of labels between time bins labeled as 'background'. To understand how this impacts performance, we computed frame error and syllable error rate with and without post-processing, as shown in *Figure 5*. We found that post-processing had little effect on the frame error (compare dashed and solid lines in *Figure 5A* top row), but that it greatly reduced the syllable error rates (bottom row). To understand this difference, we performed further analysis. We found that many of the frame errors could be attributed to disagreements between the model predictions and the ground truth annotations about the onset and offset times of syllables (see *Figure 5—figure supplement 1*). These syllable boundaries are naturally variable in the ground truth data, but such mismatches between the model predictions and the ground truth do not change the label assigned to a segment, and thus do not contribute to the syllable error rate. We also asked whether the increased syllable error rate might be due to errors that result when the model sees sequences of syllables that occur with very low probability. We were unable to find strong evidence that these infrequently-seen sequences caused the model to make errors. Rarely occurring sequences had little effect even when we limited the performance of our model by shrinking the size of the hidden state in the recurrent layer (see *Figure 5—figure supplement 2*). The results of this further analysis and the difference we observed between frame error and syllable error rate suggested to us that our post-processing corrects a small number of mislabeled frames peppered throughout the network outputs, which has a comparatively large effect on the syllable error rate.

Next we sought to understand how the hyperparameters used during training affected the small number of incorrect frames that inflate the syllable error rate. We focused on two key hyperparameters we considered most likely to affect syllable error rate: the size of windows from spectrograms shown to the network (measured in the number of time bins), and the size of the hidden state in the recurrent layer. The window size determines the context the network sees, while the hidden state size determines the network's capacity to integrate contextual information across time steps. We ran further experiments using a range of values for both hyperparameters to determine how they impact performance. In all cases, we saw that both hyperparameters had little effect on frame error (top row in *Figure 5B and C*) but a large effect on syllable error rate (bottom row in *Figure 5B and C*). This difference between metrics is again consistent with the idea that the main contributor to the syllable error rate is a handful of frame errors scattered across the network outputs. These experiments also confirmed that the values we chose to obtain results in *Figures 3 and 4* were close to optimal; smaller values would have negatively impacted performance, and larger values would have yielded little or no additional gain. (There is no widely-accepted method to find truly optimal hyperparameters.) In all cases, the effect of these hyperparameters was clear when looking at the model outputs before post-processing (orange boxes in *Figure 5B and C*). We did not see any similar effect when testing other hyperparameters such as filter size (*Figure 5—figure supplement 2*) and number (*Figure 5—figure*

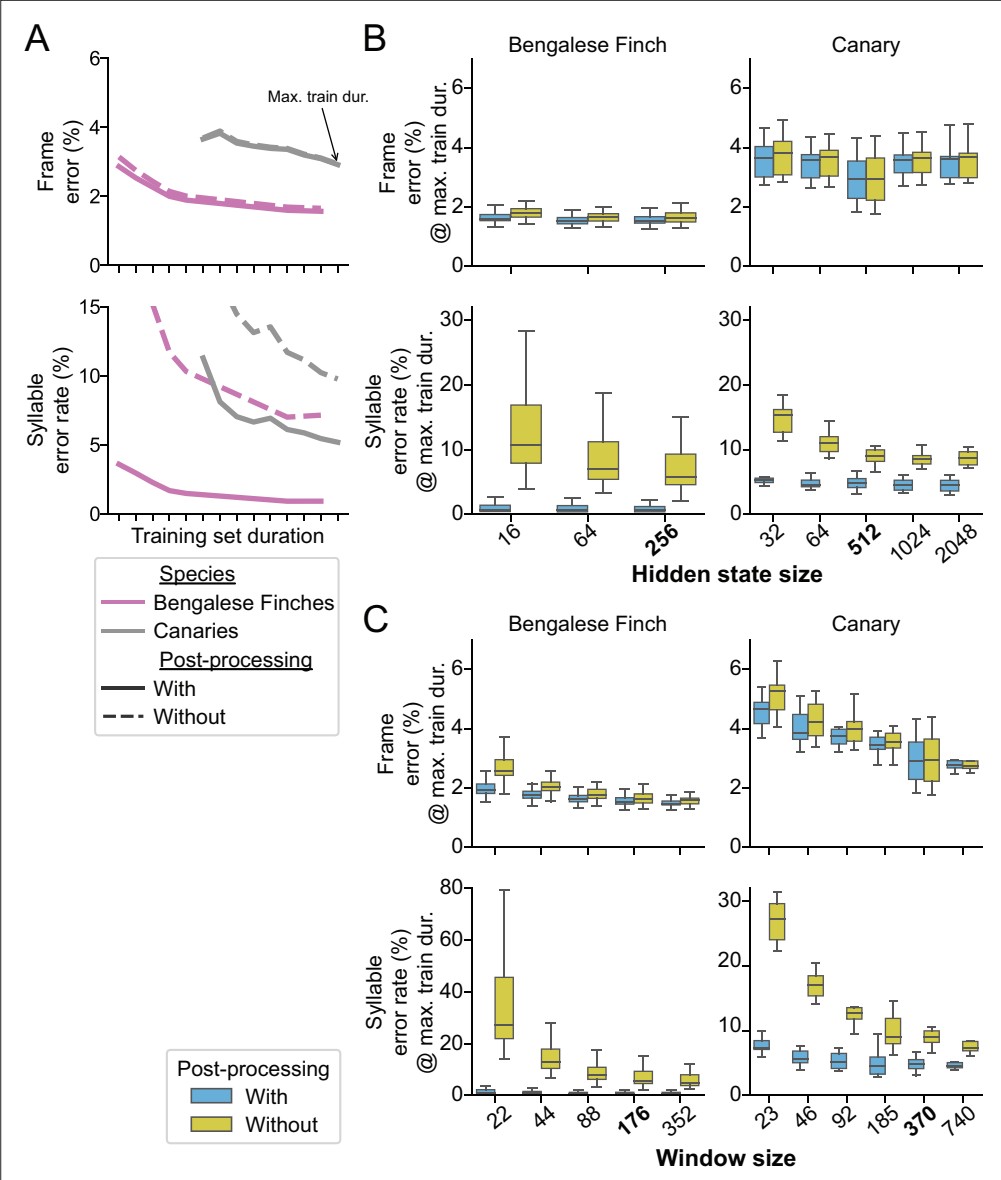

**Figure 5.** The effect of post-processing and hyperparameters on TweetyNet performance. (**A**) Mean frame error (top row) and mean syllable error rate, across all birds and training replicates, as a function of training set duration. Hue indicates species (Bengalese finches, magenta; canaries, dark gray). Line style indicates whether the metric was computed with (solid lines) or without (dashed lines) post-processing of the vectors of labeled time bins that TweetyNet produces as output. (Note solid lines are same data as *Figure 4*). (**B, C**). Performance for a range of values for two key hyperparameters: the size of windows from spectrograms shown to the network (**B**) and the size of the hidden state in the recurrent layer (**C**). Box-and-whisker plots show metrics computed at the maximum training set duration we used for the curves in A ('Max. train dur.', black arrow in A). We chose the maximum training set durations because at those metrics were closest to the asymptotic minimum approached by the learning curves. Top row of axes in both B and C shows frame error, and bottom row of axes shows syllable error rate. Blue boxes are metrics computed with post-processing transforms applied, orange boxes are error rates without those transforms. Ticks labels in boldface on axes in B and C represent the hyperparameters we used for results shown in A, and *Figures 3 and 4*.

The online version of this article includes the following source data and figure supplement(s) for figure 5:

**Source data 1.** Data used to generate line plots in *Figure 5A, B*, *Figure 5—figure supplement 3*.

**Source data 2.** Data used to generate box plots in *Figure 5B, C*, *Figure 5—figure supplement 3*.

**Figure supplement 1.** Most frame errors of trained TweetyNet models are disagreement on syllable boundaries of

*Figure 5 continued on next page*

*Figure 5 continued*

0–2 time bins.

**Figure supplement 2.** frame errors in rarely-occurring Bengalese finch sequences.

**Figure supplement 3.** Filter size experiments.

**Figure supplement 4.** Filter number experiments.

*supplement 3*). In total, these results show that our algorithm is in fact learning something about the sequences, by leveraging context from the windows that it sees and by storing information it propagates across time steps in its hidden state.

However, our simple post-processing step had a much larger effect on both error metrics, making the impact of hyperparameters difficult to see when plotted at the same scale (blue boxes in *Figure 5B and C*). Therefore, the results also demonstrate that even with well-chosen hyperparameters the network outputs contain segmenting errors that our post-processing removes. In all cases, we were able to reduce the syllable error rate by nearly an order of magnitude with post-processing. We return to this point in the discussion.

## Birdsong annotated automatically with TweetyNet replicates key behavioral findings

We next assessed performance of TweetyNet in a scenario more similar to how an experimentalist would apply our approach to their data. Specifically, we asked whether we could replicate key findings from previous behavioral experiments, using annotations predicted by TweetyNet.

### TweetyNet annotation of Bengalese finch song replicates statistics of branch points

Bengalese finch song is known to contain *branch points*, where one class of syllable can transition to two or more other classes. An example is shown in *Figure 6A and B*. *Warren et al., 2012* showed that these transition probabilities are stable across many days. We asked if we could replicate this result with automated annotation for several full days of recordings, predicted by TweetyNet models trained on a relatively small set of manually-annotated songs. To do so, we used the dataset from *Nicholson et al., 2017*, that contains recordings from 4 Bengalese finches, whose every song was manually annotated across 3–4 days. We verified that in the ground truth annotations from *Nicholson et al., 2017* we could replicate the key finding from *Warren et al., 2012*, that branch point statistics were stable across several days (Bonferroni-corrected pairwise bootstrap test, n.s. in all cases).

Before testing whether we could recover this finding from annotations predicted by TweetyNet, we first measured model performance across entire days of song. Using models trained on 10 min (for experiments in *Figures 3 and 4*), we predicted annotations for the remainder of the songs. As shown in *Figure 6C*, we found that these TweetyNet models maintained low syllable error rates when measured with entire days of song, without exhibiting large fluctuations across days. The syllable error rate ranged from 1% to 5% across 3–4 days of song from each of the four birds, comparable to rates observed in *Figure 4*. We emphasize that the days of songs we used as test sets here are much larger than those we used to benchmark models in *Figure 4*. The mean duration of these test sets was 1528 seconds (s.d. 888.6 s, i.e. 25 min mean, 14 min s.d.), in contrast to *Figure 4* where we measured syllable error rates with a fixed test set of 400 s (6 min 40 s).

Next we asked whether we could recover the behavioral findings using annotations predicted by TweetyNet. Applying the same analysis from *Warren et al., 2012*, we found that annotations predicted by TweetyNet were statistically indistinguishable from the ground truth data (Bonferroni-corrected pairwise bootstrap test, again n.s. in all cases). This can be seen by overlaying model and ground truth predictions, as in the representative example in *Figure 6D*. Summary results for all branch points on all days in all four birds are shown in *Figure 6E*, again illustrating that the probabilities estimated from predicted annotations were quite similar to those estimated from the ground truth.

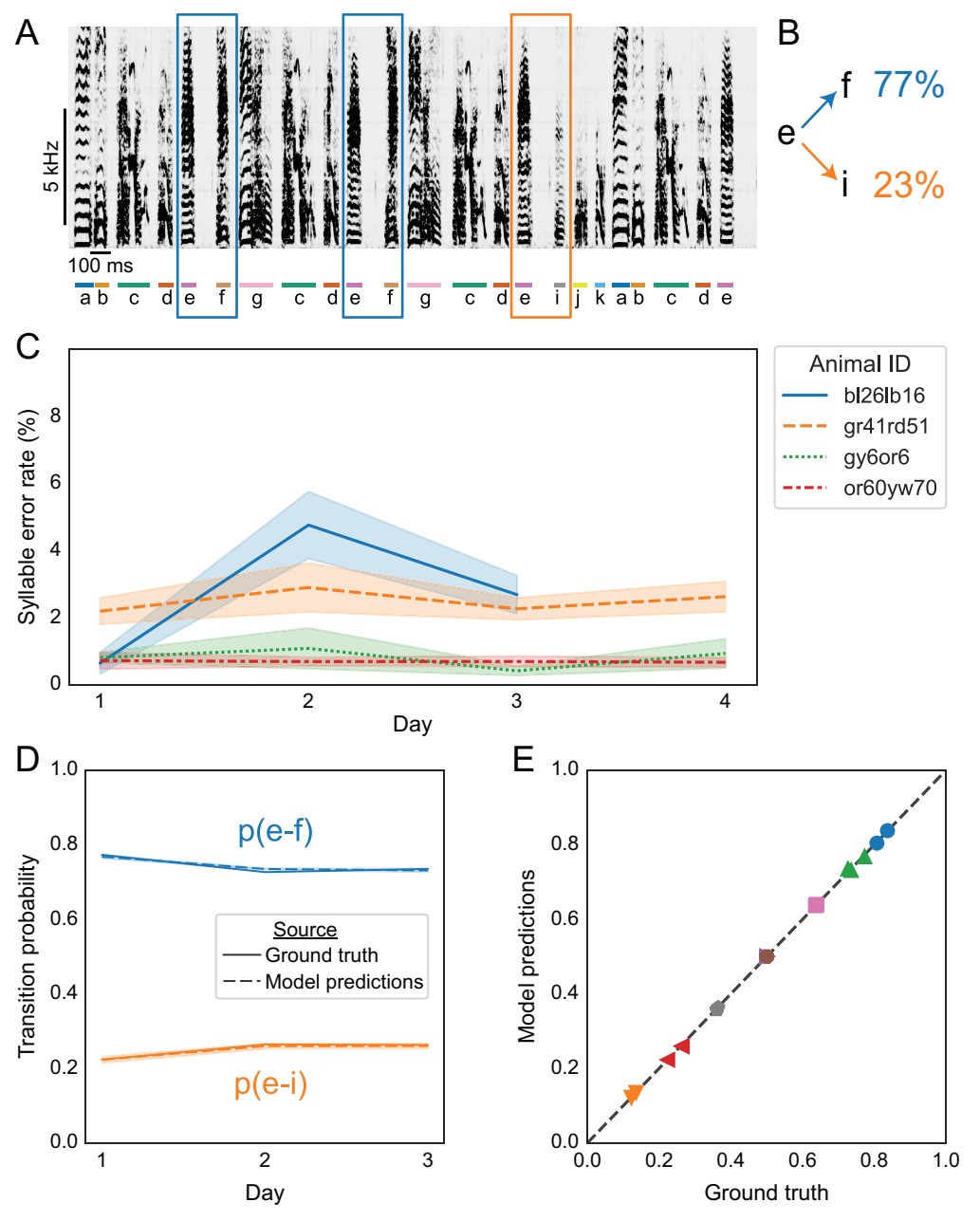

**Figure 6.** Replicating results on branch points in Bengalese finch song with annotations predicted by TweetyNet. (**A**) Representative example of a Bengalese finch song with a *branch point*: the syllable labeled 'e' can transition to either 'f', as highlighted with blue rectangles, or to 'i', as highlighted with an orange rectangle. (**B**) Transition probabilities for this branch point, computed from one day of song. (**C**) Syllable error rates per day for each bird from *Nicholson et al., 2017*. Solid line is mean and shaded area is standard deviation across 10 training replicates. Line color and style indicate individual animals. TweetyNet models were trained on 10 min of manually annotated song, a random subset drawn from data for day 1. Then syllable error rates were computed for the remaining songs from day 1, and for all songs from all other days. (**D**) Transition probabilities across days for the branch point in A and B, computed from the ground truth annotations (solid lines) and the annotations predicted by TweetyNet (dashed lines). Shaded area around dashed lines is standard deviation of the estimated probabilities, across the 10 training replicates. (**E**) Group analysis of transition. x axis is probability computed from the ground truth annotations, and the y axis is probability estimated from the predicted annotations. Dashed line is 'x = y', for reference. Each (color, marker shape) combination represents one branch point from one bird.

The online version of this article includes the following source data for figure 6:

*Figure 6 continued on next page*

Figure 6 continued

**Source data 1.** Data used to generate line plot in *Figure 6C*.

**Source data 2.** Data used to generate line plot in *Figure 6D*.

**Source data 3.** Data used to generate scatter plot in *Figure 6E*.

## TweetyNet annotation of canary song replicates statistical models of song structure

Canary songs consist of trills of repeated syllables called phrases (*Figure 1B*). *Markowitz et al., 2013* examined sequences of phrases of Waterslager canaries and found transitions with different memory depths. They showed this by describing probability distribution of transition outcomes from

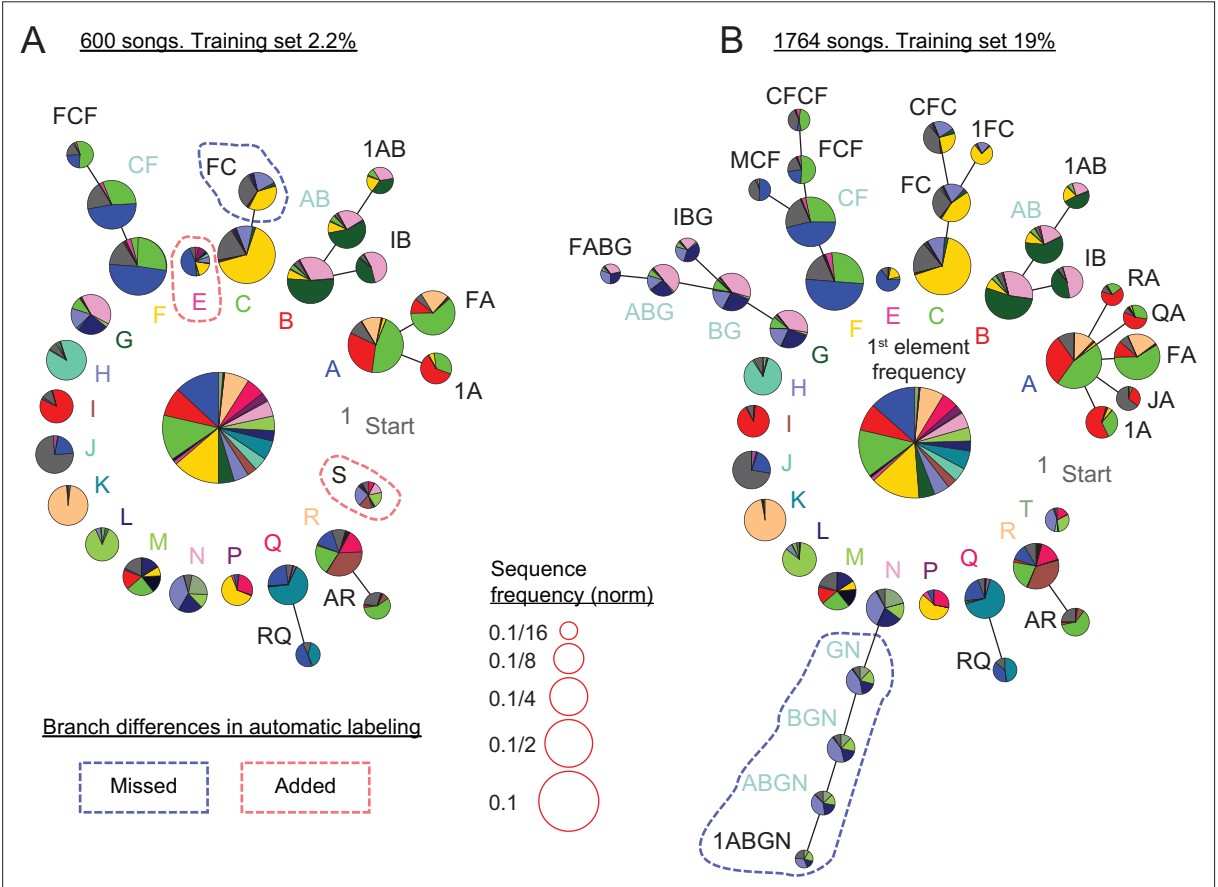

**Figure 7.** Replicating and extending results about canary syntax dependencies with annotations predicted by TweetyNet. (**A**) Long-range order found in 600 domestic canary songs annotated with human proof reader (methods, similar dataset size to *Markowitz et al., 2013*). Letters and colors indicate phrase types. Each branch terminating in a given phrase type indicates the extent to which song history impacts transition probabilities following that phrase. Each node corresponds to a phrase sequence, annotated in its title, and shows a pie chart representing the outgoing transition probabilities from that sequence (e.g. the pie '1A' shows the probabilities of phrases 'B', 'C', and 'F' which follow the phrase sequence '1→ A'). The nodes are scaled according to their frequency (legend). Nodes that can be grouped together (chunked as a sequence) without significantly reducing the power of the model are labeled with blue text. These models are built by iterative addition of nodes up the branch to represent longer Markov chains, or a transition's dependence on longer sequences of song history. A TweetyNet model was trained using 2.2% of 1,764 songs (9.5% of the data in A). The PST created from the model's predicted annotation of the entire dataset is very similar to A (see full comparison in *Figure 7—figure supplement 1*). Here, branch differences between the hand labeled and model labeld song are marked by red and blue dashed lines for added and missed branches. (**B**) PST created using all 1,764 hand labeled songs. An almost identical PST was created *without* a human proof reader from a TweetyNet model trained on 19% of the data (see full comparison in *Figure 7—figure supplement 2*).

The online version of this article includes the following figure supplement(s) for figure 7:

**Figure supplement 1.** Detailed comparison of syntax structure in 600 hand labeled or TweetyNet-labeled canary songs.

**Figure supplement 2.** Detailed comparison of syntax structure in 1764 hand labeled or TweetyNet-labeled canary songs.

certain phrases by Markov chains with variable lengths. This syntax structure is captured parsimoniously by probabilistic suffix trees (PST) (*Ron et al., 1996*). The root node in these graphical models, appearing in the middle of *Figure 7A and B*, represents the zero-order Markov, or base rate, frequencies of the different phrases, labeled in different colors and letters. Each branch, emanating from the colored letters in *Figure 7*, represents the set of Markov chains that end in the specific phrase type designated by that label. For example, the 'A' branch in *Figure 7A* includes the first order Markov model 'A' and the second order Markov chains 'FA' and '1A' representing the second order dependence of the transition from phrase 'A'.

We asked if we could replicate findings about canary song syntax in a different strain of canaries using a TweetyNet model trained on a small manually annotated dataset. *Figure 7* demonstrates that annotations predicted by TweetyNet had sufficient accuracy on domestic canary song to extract its long-range order. In these figures, we set parameters of the PST estimation algorithm to derive the deepest syntax structure possible without overfitting, following the approach of *Markowitz et al., 2013* that used about 600 hand-annotated songs of Waterslager canaries. In this example, using 2.2% of the data set, about 40 songs, to train a TweetyNet model and predict the rest of the data reveals the deep structures shown in *Figure 7A*, comparable to using 600 hand annotated songs of the same bird. With more training data, Tweetynet's accu-

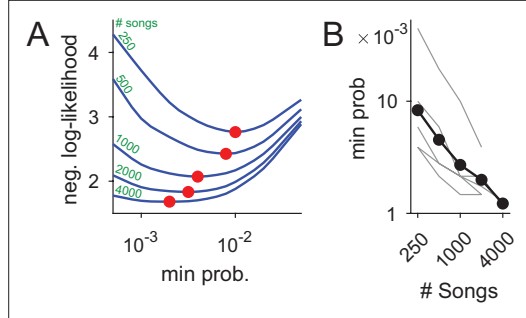

**Figure 8.** Using datasets more than five times larger than previously explored increases statistical power and the precision of syntax models. (**A**) Ten-fold cross validation is used in selection of the minimal node probability for the PSTs (x-axis). Lines show the mean negative log-likelihood of test set data estimated by PSTs in 10 repetitions (methods). Curves are calculated for datasets that are sub sampled from about 5000 songs. Red dots show minimal values - the optimum for building the PSTs. (**B**) The decrease in optimal minimal node probability (y-axis, red dots in panel A) for increasing dataset sizes (x-axis) is plotted in gray lines for six birds. The average across animals is shown in black dots and line.

The online version of this article includes the following source data for figure 8:

**Source data 1.** Data used to generate lines in *Figure 8A*.

**Source data 2.** Data used to generate dots in *Figure 8A*.

**Source data 3.** Data used to generate lines in *Figure 8B*.

racy improves as does the statistical strength of the syntax model. In *Figure 7B* a TweetyNet model was trained on 19% of the data, about 340 songs, and predicted the rest of the data. The resulting syntax model can be elaborated to greater depth without overfitting. To validate this deeper model, we compared it with a PST fit to all 1764 songs from the same bird, manually annotated, and found that both PSTs were very similar (*Figure 7B*).

In sum, we find that TweetyNet, trained on a small sample of canary song, is accurate enough to automatically derive the deep structure that has formed the basis of recent studies (*Markowitz et al., 2013*; *Cohen et al., 2020*).

## Larger data sets of annotated canary song add details and limit the memory of the syntax structure

The increase in syntax detail, presented in *Figure 7B*, is possible because more rare nodes can be added to the PST without over-fitting the data. Formally, the PST precision increase in larger data sets is defined by the decrease in minimal node frequency allowed in the process of building PST models (*Figure 8*), as measured in model cross validation (see Materials and methods). In our data set, we find an almost linear relation between the number of songs and this measure of precision—close to a tenfold precision improvement.

In *Figure 7B*, this increased precision allowed reliably adding longer branches to the PST to represent longer Markov chains (in comparison to *Figure 7A*). In this example, using a dataset three times larger revealed a 5-deep branch that initiates with the beginning of song ('1ABGN'), suggestive of a potential global time-in-song dependency of that transition. The PST in *Figure 7B* also has branches

that did not 'grow' compared to *Figure 7A* when more songs were analyzed (e.g. the 'B', 'Q', and 'R' branches), indicating a potential cutoff of memory depth that is crucial in studying the neural mechanisms of song sequence generation.

The data sets used in *Figures 7 and 8*, are about 10 times larger than previous studies. To ascertain the accuracy of the syntax models, in creating the data sets we manually proofread annotations predicted by TweetyNet (see 'Annotation of canary song' in Materials and methods). Across five different human proof readers, we compared the time required to manually annotate canary song with the proof-reading time, and found that using TweetyNet saved 95–97.5% of the labor.

Taken as a whole, results in this section show that TweetyNet makes high-throughput automated annotation of behavioral experiments possible, greatly reducing labor while scaling up the amount of data that can be analyzed by orders of magnitude.

## Very rare, hard-to-classify vocal behaviors can cause TweetyNet to introduce errors

Songbird species vary in the degree to which the elements of their song can be categorized into a set of discrete classes (*Thompson et al., 2012*; *Sainburg et al., 2020*). Even for species where expert annotators can readily define such a set, there will occasionally be periods in song where it is unclear how to classify syllables. Here, we provide examples of these rare cases to illustrate how even a well-trained TweetyNet model can introduce errors in annotation when the behavior itself cannot be cleanly categorized. The examples we present in *Figure 9* are from canaries, simply because their song can be so highly varied. As these examples illustrate, predictions of TweetyNet models are well-behaved when faced with rare variants, assigning high probability to the most relevant labels, not to completely unrelated classes of syllables. We emphasize that any standard supervised machine learning model that assigns only a single label to each segment will be vulnerable to introducing errors like these. Such errors raise questions about whether and when birdsong can be categorized into discrete syllable classes, questions that are brought back into focus by methods like ours that automate the process. As we will now discuss, we see several ways in which future work can address these questions.

## Discussion

Annotating birdsong at the level of syllables makes it possible to answer questions about the syntax governing this learned sequential behavior (*Berwick et al., 2011*). Annotating syllables also makes it possible describe them in physical units like pitch and amplitude that researchers can directly link to muscular and neural activity (*Sober et al., 2008*; *Wohlgemuth et al., 2010*). However, for many species of songbirds, analyses at the syllable level still require labor-intensive, time-consuming manual annotation. There is a clear need for a method that can automate annotation across individuals and species, without requiring cleanly segmented audio, and without requiring researchers to carefully tune and validate multiple statistical models. To meet this need, we developed a neural network, TweetyNet (*Figure 2*): a single model trained end-to-end that learns directly from spectrograms how to automatically annotate each bird's unique song. TweetyNet is deliberately designed to avoid dividing annotation into separate steps of segmentation and labeling, and it leverages the strengths of deep learning models to learn features for classification from the training data, instead of relying on pre-defined engineered features. We showed that our approach mitigates issues that result from the assumption that audio can be cleanly segmented into syllables (*Figure 3*). TweetyNet performs comparably to a carefully tuned Support Vector Machine model operating on pre-defined features extracted from manually cleaned, perfectly segmented audio (*Figure 3*). This result might suggest that an alternative to our approach would be to improve the audio segmentation step (e.g. with an alternative algorithm *Tchernichovski et al., 2000*) and to use a state-of-the-art non-neural network model (such as XGBoost *Chen and Guestrin, 2016*). Because such approaches lack the flexibility and expressiveness of deep learning models, we believe they will still require additional tuning our method avoids. For example, one could add classes for background noise to such models, but this would likely require additional steps to deal with class imbalance. Our model and problem formulation *requires* adding an additional 'background' class, which results in a more general solution (in much the

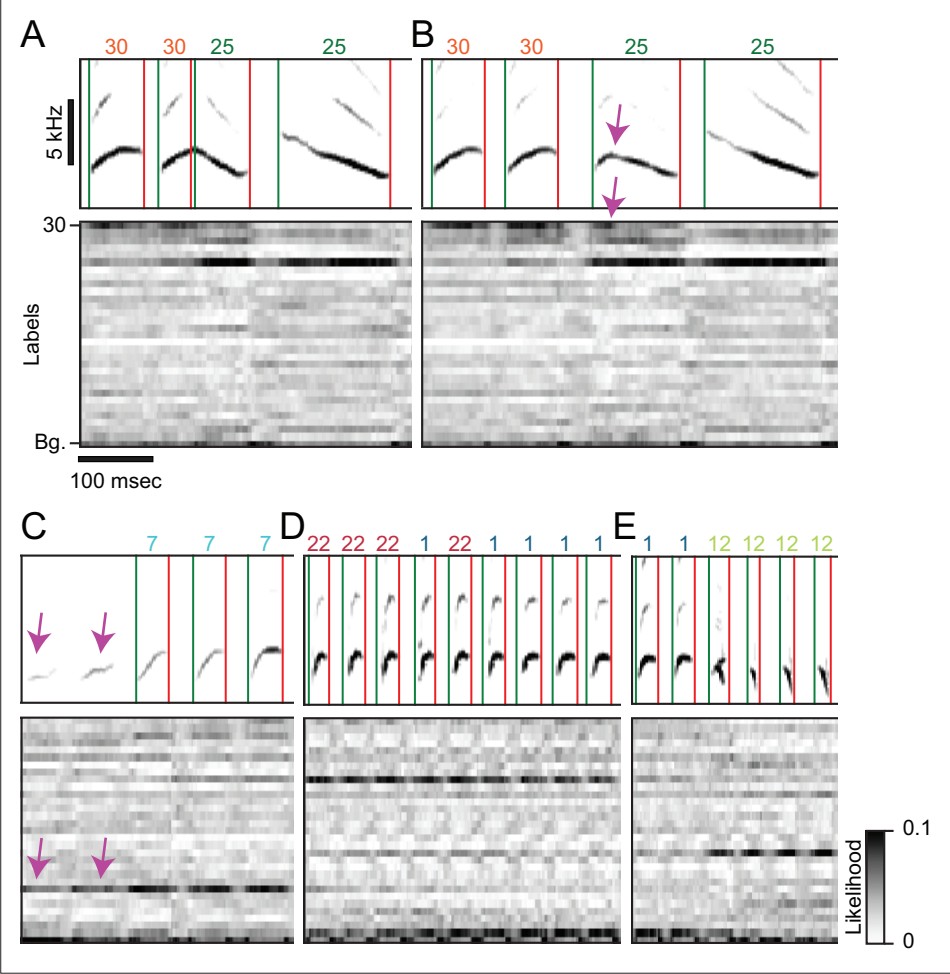

**Figure 9.** Rare variants of canary song introduce segmentation and annotation errors. (**A-E**) Spectrograms on top of the time-aligned likelihood (gray scale) assigned by a well-trained TweetyNet model to each of the labels (y-axis, 30 syllable types and the tag '*Bg.*' for the background segments). Green and red vertical lines and numbers on top of the spectrograms mark the onset, offset, and labels predicted by the model. (**A,B**) Canary phrase transitions can contain a vocalization resembling the two flanking syllables fused together. A TweetyNet model trained to split this vocalization performed very well (**A**) but failed in a rare variant (**B**). The network output highlights a general property: TweetyNet assigned high likelihood to the same flanking syllable types and not to irrelevant syllables. (**C**) Syllables produced soft, weak, and acoustically imprecise at the onset of some canary phrases are mostly captured very well by TweetyNet but, on rare occasions, can be missed. In this example the model assigned high likelihood to the correct label but higher to the background. (**D**) Some human annotators, called 'splitters', define more syllable classes. Others, the 'lumpers', group acoustically-diverse vocalizations under the same label. TweetyNet models trained on acoustically-close classes assign high likelihood to both labels and, on rare occasions, flip between them. This example demonstrates that TweetyNet does not use the a-priori knowledge of syllable repeats hierarchically-forming canary phrases. (**E**) Canaries can simultaneously produce two notes from their two bronchi. This occurs in phrase transitions and the spectrogram of the resulting vocalization resembles an overlay of flanking syllables. While the network output shows high likelihood for both syllables the algorithm is forced to choose just one.

same way that object detection models require and benefit from the addition of a background class *Scheirer et al., 2013*).

A natural question raised by our results is how TweetyNet integrates the local acoustic structure within a few time bins of a spectrogram and the global structure of syllable sequences within spectrogram windows. To answer this, and gain a deeper understanding of our approach, we carried out experiments varying two key hyperparameters. The first was the size of spectrogram windows shown to the network, which determines how much context the network sees, and the second was the size

of the hidden state in the recurrent layer, which determines the network's capacity to integrate information across time steps. The experiments demonstrated that TweetyNet performance depends on well-chosen values for both hyperparameters (*Figure 5*). These empirical results support the idea that TweetyNet learns to recognize local features seen in syllables *and* learns to leverage sequential information across a spectrogram window.

In addition, we showed our approach performs well across multiple individuals from two different species of songbird, Bengalese finches and canaries (*Figure 4*). We demonstrated that using automated annotations produced by TweetyNet, we could replicate key findings from long-term behavioral experiments about the syntax of Bengalese finch (*Figure 6*) and canary song (*Figures 7 and 8*). Overall, these results show that our deep learning-based approach offers a robust, general method for automated annotation of birdsong at the syllable level.

## Ideas and speculation

Our results open up avenues for future research in two directions: development of neural network algorithms, and applications of those algorithms, once developed. Regarding algorithm development, we suggest that future work should test whether networks can be trained to better learn to segment *without* post-processing. The experiments in *Figure 5* suggested that the post-processing we applied improves performance by correcting a small number of mislabeled time bins that cause a large increase in syllable error rate. From this, and from previous related work (*Lea et al., 2017*), our sense is that a logical next step will be to incorporate the syllable error rate into the loss function, minimizing it directly. This would require some modifications to our approach, but may prove more effective than testing different network architectures.

Another important question for future work is: when it is appropriate to apply supervised learning algorithms to vocalizations, like ours and related object detection-based models (*Coffey et al., 2019*; *Fonseca et al., 2021*), and when should these algorithms be combined or even replaced with unsupervised algorithms. Recently developed unsupervised models learn helpful, often simpler, representations of birdsong and other animal vocalizations (*Goffinet et al., 2021*; *Sainburg et al., 2019*, *Sainburg et al., 2020*). These advances and the advantages of methods like TweetyNet are not mutually exclusive, and can be integrated in different ways depending on the needs of researchers. For example, a TweetyNet model can serve as a front-end that finds and classifies segments, which are then passed to an unsupervised model. In addition to annotating syllables, we suggest future work consider two other levels of classification. The first would use TweetyNet to segment audio into just two classes: 'vocalization' and 'non-vocalization' periods. Treating segmentation as a binary classification problem in this way would make it possible to extend our approach to vocalizations that are not easily categorized into discrete labels: juvenile birdsong, bat calls (*Prat et al., 2017*), and rodent USVs (*Tachibana et al., 2020*), for example. Another level of classification consists of automatically annotating higher-level structures in song such as motifs, phrases (*Markowitz et al., 2013*) or chunks (*Takahasi et al., 2010*; *Kakishita et al., 2008*). TweetyNet could annotate these explicitly defined higher level structures, that would then be passed to downstream unsupervised models designed for tasks like similarity measurement (e.g. *Goffinet et al., 2021*; *Sainburg et al., 2019*, *Sainburg et al., 2020*). A second way that supervised and unsupervised algorithms could be combined would be to reverse the order, and use the unsupervised model as a front end. For example, models like those of *Sainburg et al., 2020* could be used to automatically generate a candidate set of syllable classes from a relatively small dataset of cleanly segmented song. A researcher would visually inspect and validate these candidate classes, and once validated, use them with TweetyNet to bootstrap annotation of a much larger dataset.

Lastly, we speculate on use of trained TweetyNet models to measure uncertainty and similarity. These measures can be estimated using either the probabilities that TweetyNet produces as outputs, or with so-called 'activations' within layers of the network that are elicited by feeding inputs through it. The output probabilities can serve as a metric in syllable space. For example, when predicting new annotations, researchers could use output probabilities from TweetyNet to flag less confident predictions for subsequent human inspection and post-processing. As shown in *Figure 9D*, this approach can highlight rare song variants and may also help annotators identify edge cases where they have defined syllable classes that are too similar to each other. More generally, a researcher could use a TweetyNet model trained on a single adult's song to obtain an estimate of any other song's similarity

to it, such as the adult's offspring or even recordings of the same adult's song as a juvenile. This could be done using the output probabilities, or activations within the network. Activations in trained TweetyNet models could also be used to assess the output of unsupervised models that generate vocalizations (*Sainburg et al., 2019*), analogous to similar approaches in computer vision (*Salimans et al., 2016*; *Heusel et al., 2017*).

## Conclusion

The family of songbirds that learns by imitation consists of over 4500 species. Some of these singers, such as the canary, produce songs that are much too complex to be automatically annotated with existing methods, and for these complex singers little is known about the syntax structure and organization of song. The results we present suggest that our approach makes automated syllable-level annotation for many of these species possible. By sharing trained models, tutorials, data, and a library for benchmarking models, we also establish a strong baseline for work building upon ours. We are confident our method enables songbird researchers to automate annotation of very large datasets of entire bouts of song, required for analyses that address central questions of sensorimotor learning.

## Materials and methods

### Data preparation

#### Segmenting audio files into syllables

##### Algorithm

For Bengalese finch song, we applied a widely-used simple algorithm to segment audio into syllables, as described in the Introduction and shown in *Figure 1A*. The first step of this algorithm consists of finding all periods (colored line segments, middle and bottom axes of *Figure 1A*) where the amplitude of song stays above some threshold (dashed horizontal line on bottom axes of *Figure 1A*). The resulting segments are further cleaned up using two more parameters. Any periods between segments that are shorter than a minimum silent interval are removed, merging any syllables neighboring those intervals, and then finally any remaining segments shorter than a minimum syllable duration are removed. We used the implementation of this audio segmenting algorithm in the evfuncs tool (*Nicholson, 2021c*), that correctly replicates segmentation of the (*Nicholson et al., 2017*) dataset, which was segmented using Matlab code developed for previous papers (see for example *Tumer and Brainard, 2007*).

#### Estimating segmenting parameters for canary song

As we state in the Introduction, the same algorithm cannot be applied to canary song. In spite of this, we apply the algorithm to canary song for results in *Figure 3*, to make very clear the issues that would results from relying on it. To estimate parameters that would produce the least amount of errors when segmenting canary song with this algorithm, we wrote a script that found the following for each bird's song: (1) the median amplitude at all syllable onsets and offsets in the ground truth data with segmentation adjusted by human annotators, (2) the 10th percentile of syllable durations, (3) and the 0.1th percentile of silent intervals between syllables. We visually inspected the distributions of these values extracted from all segments, with our estimated segmenting parameters superimposed, to validate that we would not artificially create a very large number of errors by using the parameters we found with this script.

#### Annotation of Bengalese finch song

Experiments in *Figure 4* included song from four birds in the 'BirdsongRecognition' dataset (*Koumura, 2016*). The models in the original study were designed to annotate specific sequences within song, as described in their methods. The goal of our model is annotate entire bouts of song. To use that dataset in our experiments, we needed to fully annotate all bouts of song. If we did not label all syllables, then our model would be unfairly penalized when it correctly annotated syllables that were present in the original dataset, but were not annotated. Two of the authors (Cohen and Nicholson) fully annotated the song from four of the birds, employing the same GUI application used to annotate canary song. The vast majority of syllables that we labeled were the low-frequency, high-entropy 'introduction' notes that occur at the beginning of some song bouts in varying numbers, that are often

ignored during analysis of zebra finch and Bengalese finch song. For the handful of cases where other syllables were not labeled, we chose from among the classes present in the already-annotated data to assign labels to these. In some very rare cases, we found syllables where the category was not clear, similar to the cases we describe for canary song in *Figure 9*. We chose to assign a separate class to these and remove song bouts containing theses classes from both the training and test sets. As we acknowledge in *Figure 9* Discussion, an inability to handle edge cases like these is a limitation of any standard supervised learning algorithm like ours, that operates at the level of syllables. We removed these cases so that we could be sure that benchmarking results accurately reflected how well the model performed on well-classified syllables.

## Annotation of canary song
### Bootstrapping annotation with TweetyNet
In this manuscript, we used annotated domestic canary datasets an order of magnitude larger than previously published. To create these datasets we used TweetyNet followed by manual proofreading of its results. This process, described below, allowed 'bootstrapping' TweetyNet's performance. Song syllables were segmented and annotated in a semi-automatic process:

- A set of 100 songs was manually segmented and annotated using a GUI developed in-house (*Cohen, 2022*). This set was chosen to include all potential syllable types as well as cage noises.
- The manually labeled set was used to train TweetyNet (*Nicholson, 2022*).
- In both the training phase of TweetyNet and the prediction phase for new annotations, data is fed to TweetyNet in segments of 1 second and TweetyNet's output is the most likely label for each 2.7 ms time bin in the recording.
- The trained algorithm annotated the rest of the data and its results were manually verified and corrected.

### Assuring the identity and separation of syllable classes
The manual steps in the pipeline described above can still miss rare syllable types or mislabel syllables into the wrong classes because of the human annotator's mistake or because some annotators are more likely to lump or split syllable classes. To address this potential variability in canaries, where each bird can have as many as 50 different syllables, we made sure two annotators agree on the definition of the syllable classes. Then, to make sure that the syllable classes are well separated, all the spectrograms of every instance of every syllable, as segmented in the previous section, were zero-padded to the same duration. An outlier detection algorithm (IsolationForest) was used to flag and re-check potential mislabeled syllables or previously unidentified syllable classes.

### Segmenting annotated phrases of Waterslager canaries
In *Figure 1—figure supplement 2* we include data from waterslager canaries, available from a previous project in the Gardner lab (*Markowitz et al., 2013*). To include this data, we needed to break annotated phrase segments into syllable segments. Songs were previously segmented into phrases, trilled repetitions of syllables, and not to individual syllables. In each segmented phrase, we separated vocalization and noise fluctuations between vocalizations by fitting a two-state hidden Markov model with Gaussian emission functions to the acoustic signal. Putative syllable segments produced by this procedure were proofread and manually corrected using a GUI developed in-house.

### Generating spectrograms
Spectrograms were generated from audio files using custom Numpy (Bengalese finch) or Matlab (canary) code. For Bengalese finches, the code we used to generate spectrograms is built into the vak library. For canaries, the code we used to generate spectrograms can be found here (*Markowitz, 2022b*).

All spectrograms for song from a given species were created with the same parameters, such as the number of samples in the window for the Fast Fourier Transform (NFFT). For Bengalese finch song, we used NFFT = 512 with a step size of 64. For canaries we used NFFT = 1024 with a step size of 119. This produced spectrograms with a time bin size of 1ms for Bengalese finches, and 2.7ms for canaries.

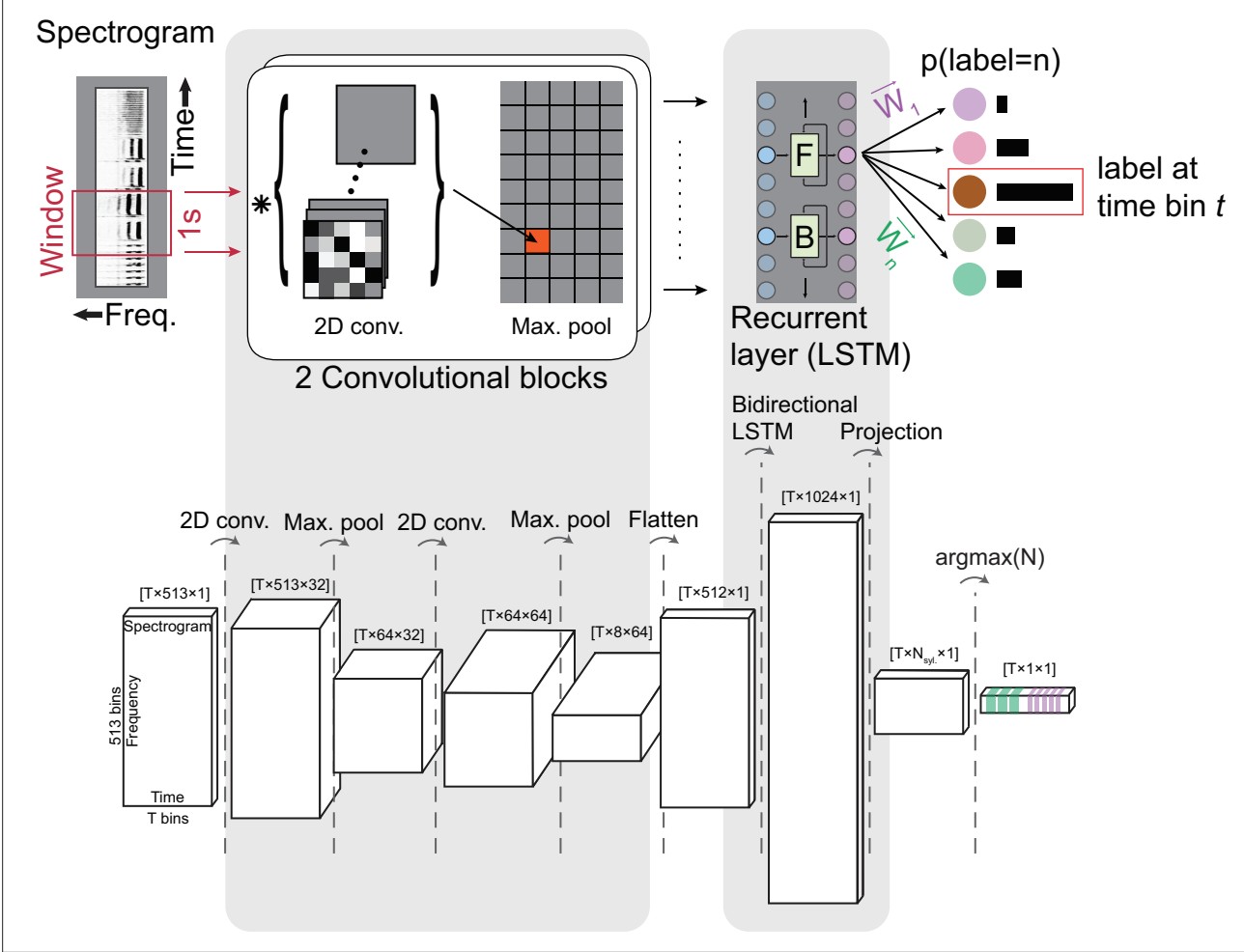

**Figure 10.** TweetyNet architecture and tensor shapes resulting from each operation in the network.

We chose spectrogram parameters such that the duration of a time bin was just smaller than the shortest duration silent gaps between syllables. A larger bin size would have prevented our model from producing correct segments, whenever one of the true silent gaps was shorter than our bin size. In initial studies we experimented with even smaller bin sizes, but found that the network tended to over-segment. Future work could compensate for this oversegmenting by modifying the loss function as we point out in the Discussion.

## Neural network architecture

Here, we provide a detailed description of the TweetyNet neural network architecture. The architecture that we develop is most directly related to those that have been used for event detection in audio and video (*Böck and Schedl, 2012*; *Parascandolo et al., 2016*) and for phoneme classification and sequence labeling (*Graves and Schmidhuber, 2005*; *Graves, 2012*). It is also somewhat similar to deep network models for speech recognition, but a crucial difference is that state-of-the-art models in that area map directly from sequences of acoustic features to sequences of words (*Graves et al., 2006*). The success of these state-of-the-art models is attributed to the fact that they learn this mapping from speech to text, *avoiding* the intermediate step of classifying each frame of audio, as has previously been shown (*Graves, 2012*). In other words, they avoid the problem of classifying every frame that we set out to solve.

As described in the introduction, the network takes as input batches of windows from a spectrogram (c.f. top of *Figure 2*) and produces as output a tensor of size ($m$ batches $\times c$ classes $\times t$ time bins). In *Figure 10* we show how networks blocks in that figure above relate to the shapes of tensors, and

how those shapes change as the network maps input to output. We refer to this as we give specific parameters here. Note that we leave out the batch dimension in this description.

## Convolutional blocks

The spectrogram window passes through two standard convolutional blocks, each of which consists of a convolutional layer and a max pooling layer. We use the standard term 'block' to refer to a layer that performs several operations on its inputs. For a convolutional block, the first operation is the convolution operation itself ('2D conv.' in *Figures 2 and 10*), as it is defined in the deep learning literature. This cross-correlation-like operation (asterisk in *Figures 2 and 10*) between the spectrogram window and the learned filters (greyscale boxes in *Figures 2 and 10*) produces a set of feature maps. In both convolutional blocks, we use filters of size (5 × 5), with a stride of 1. The first block contains 32 filters and the second contains 64, as shown in *Figure 10*. We pad the feature maps so that they are the same size as the input. For a spectrogram window of size (176 time bins x 513 frequency bins), as we use for Bengalese finch song, this would produce an output of (176 × 513 x 32) from the first convolution, similar to what is shown in *Figure 10*.

A key advantage of convolutional layers is that they enable 'weight sharing', that is, the relatively small number of parameters in each filter is applied to the input via the convolution operation, instead of needing to have weights for each dimension of the input (e.g. each pixel in an image). We used a full weight sharing scheme, meaning that each filter was cross-correlated with the entire input window. Previous work has tested whether performance on tasks related to ours, such as speech recognition, might be improved by alternate weight-sharing schemes, for instance by restricting filters to specific frequency ranges within a spectrograms. One previous study did report a benefit for this limited weight-sharing scheme applied to automatic speech recognition (*Abdel-Hamid et al., 2014*). However, this approach has not been widely adopted, and to the best of our knowledge, the common 'full' weight-sharing approach that we employ here is used by all state-of-the-art models for speech recognition for example, (*Amodei et al., 2016*), as well as the event detection models that we based our architecture on (*Parascandolo et al., 2016*; *Lea et al., 2017*).

## Max pooling layer

In both convolutional blocks, we followed the convolutional layer by a max pooling layer. The max pooling operation ('Pooling' in *Figure 2*) downsamples feature maps by sliding a window across the input (orange bin in *Figure 2*) and at each point keeping only the element with the maximum value within the window. We used a window size of (8 × 1) and a stride of (8, 1), with windows oriented so that the side of size one and the stride of size one were in the temporal dimension. Both the shape and stride were meant to avoid downsampling in the temporal dimension, under the assumption that it was important to retain this information. Applying a max pooling operation with these parameters to the first set of convolutional maps with size (176 × 513 x 32) produces an output tensor with size (176 × 64 x 32) as shown in as shown in *Figure 10*.

The max pooling operation is widely used in networks for related tasks like automatic speech recognition and audio event detection. Previous work has not found any benefit to alternative pooling operations such as stochastic pooling (*Sainath et al., 2013a*) and alternatives have not been widely adopted. To our knowledge most related work also adopts our approach of not down-sampling in the temporal dimension, and studies have not found any advantage when using larger strides in the temporal dimension (*Sainath et al., 2013a*).

## Recurrent layer

The output of the second convolutional block passes through a recurrent layer made up of LSTM units. Before passing it into the recurrent layer, we stack the feature maps: e.g. an output of (176 time bins x eight down-sampled frequency bins x 64 feature maps) becomes (176 time bins x 512 features) (indicated as "Flatten" in *Figure 10*). We specifically use a bidirectional LSTM, meaning the recurrent network processes the input in both the forward and backward direction. By default we set the size of the hidden state in the network equal to the 'features' dimension, and based on experiments in *Figure 5* this appears to be a reasonable default. The matrix of hidden states for all time steps become the output that we feed into the final layer. We adopt the standard practice of concatenating

the hidden states from running the sequence in the forward and backward directions, doubling its size.

## Linear layer

The final layer in TweetyNet is a linear projection ($\vec{W}_{t,s}$, purple matrix in *Figure 2*) of the recurrent layer's output onto the different syllable classes, $s = 1...n$, resulting in a vector of $n$ syllable-similarity scores for each spectrogram time bin $t$. The number of classes, $n$, is predetermined by the user. To segment syllables, the bin-wise syllable-similarity scores are first used to select a single syllable class per time bin by choosing the label with the highest syllable-similarity score. Since similarity scores can be normalized, this is akin to maximum a-posteriori (MAP) label selection. Then, the labeled time bins are used to separate continuous song segments from no-song segments and to annotate each song-segment with a single label using majority decision across time bins in that segment.

## Training and benchmarking

### Input data transformations

#### Windows

As stated above, the input to the network consists of spectrogram windows. We used a window size of 176 time bins for Bengalese finch song and 370 time bins for canary song, with the exception of experiments testing the impact of this hyperparameter in *Figure 5*.

#### Vectors of labeled time bins

We formulate annotation as a supervised learning problem where each spectrogram window $x$ has a corresponding vector of labeled time bins $y$, and our goal is to train the network $f$ to correctly map a window to this vector of labeled time bins, $f(x) \longrightarrow y$. These vectors are generated dynamically by our software from annotations consisting of segment labels and their onset and offset times. Each element in the vector $y$ contains an integer label $c_t$ corresponding to the syllable class $c$ in that time bin of the spectrogram window $x_t$. To this set of class labels, we add an additional class for the 'background' time bins that result naturally from gaps between annotated segments ('Bg.' in *Figure 2*). This 'background' class will include brief quiet periods between syllables, as well as any other longer periods left unlabeled by the annotator that may contain non-song bird calls and background noises.

#### Batches of (window, labeled time bin vector) pairs

During training, we randomly grab a batch of (window, labeled time bin vector) pairs $(x, y)$ from all possible windows in the dataset X. To achieve this, we developed a torch Dataset class that represents all such possible windows paired with the corresponding vector of labeled timebins $(x_i, y_i)$. The class tracks which windows the network has already seen during any epoch (iteration through the entire data set), ensuring that we avoid repeating the same windows during training, which could have encourage the network to memorize the training data.

The choice to randomly present windows also acts as a form of data augmentation that encourages the network to exhibit translation invariance. That is, because the network sees very similar sequences repeatedly, but those sequences are randomly shifted forward or backward slightly in time, it learns to correctly classify all time bins in a window regardless of how the window is presented to the model.

#### Normalization

Normalization is a standard practice that improves optimization of machine learning models, but is not always necessary for neural networks.

For Bengalese finch song, we normalized spectrograms; more precisely we standardized by finding the mean μ and standard deviation $\sigma$ of every frequency bin across all spectrograms in the training set, and then for every window $x$ we subtracted off the mean and divided by the standard deviation: $x_{\mathrm{normalized}} = \frac{x-\mu}{\sigma}$. Note that we achieved this with a SpectrogramScaler class built into the vak library rather than pre-processing with a script. For canary song we did not apply this normalization, and left the spectrograms as processed by the Matlab code referenced above. We did not systematically asses how normalization impacted performance.

## Spectrogram thresholding

We did not apply any thresholding to spectrograms, as is often done when visualizing them to increase contrast between sounds of interest and often quieter background noise. In preliminary experiments, we did test the effect of thresholding spectrograms, setting any value of the power spectrum less than the specified threshold to zero. However, we found that this led to a slight increase in error rates, and also made training more unstable. Our best guess for this effect of thresholding is that it produces abrupt, large magnitude changes in values in the spectrogram that may affect the gradient computed during training.

## Metrics

We define the metrics we use before describing our training methods, since our methods depend on these metrics. We measured performance of TweetyNet with two metrics.

### Frame error

The first is the frame error, that simply measures for each acoustic frame (in our case, each time bin in a spectrogram) whether the predicted label matches the ground truth label. Hence the range of the frame error is between 0 and 1, that is can be stated as a percent, and gives an intuitive measure of a model's overall performance. Previous work on supervised sequence labeling, including bidirectional-LSTM architectures similar to ours, has used this metric (*Graves, 2012*; *Graves and Schmidhuber, 2005*).

### Syllable error rate

The second metric we used is commonly called the word error rate in the speech recognition literature, and here we call it the syllable error rate. Because the syllable error rate is key to our results, we define it here, as shown in *Equation 1*.

$$
\begin{aligned}
\text{Syllable Error rate} \quad &= \frac{\text{Edit distance(reference sequence, predicted)}}{\text{Length(reference sequence)}} \\
&= \frac{\text{Substitutions + Insertions + Deletions}}{\text{Length(reference sequence)}}
\end{aligned}
\tag{1}
$$

This metric is an edit distance, that counts the number of edits (insertions, deletions, and substitutions) needed to correct a predicted sequence so it matches the ground-truth ('reference') sequence. A common algorithm used to compute the number of edits is the Levenshtein distance that we use here. The edit distance is normalized by the length of the ground truth sequence, to make it possible to compare between sequences of different lengths.

## Training

We trained all models using the Adam optimizer (*Kingma and Ba, 2014*) with a learning rate of 0.001, and other hyperparameters set to the defaults in the torch library: $(\beta_1, \beta_2 = (0.9, 0.999), \epsilon = 1e - 08, \text{weight decay} = 0.0, \text{amsgrad} = \texttt{False})$.

For all experiments, we used a batch size of 8. We specify a number of epochs in our configuration files (an epoch is one iteration through the entire training dataset) but in practice we found that the number of windows is so large that we did not complete one entire epoch of training before network performance on the validation set met the criteria for early stopping, as described in the next paragraph.

### Early stopping

To mitigate the tendency of neural networks to overfit, we employed early stopping. Error rates are measured on a validation set every val_step training steps, and training stops early if these error rates do not decrease after patience consecutive validation steps, where val_step and patience are option values declared by a user in configuration files for the vak library. We chose to specify validation in terms of a global step instead of epoch, because as just stated the size of the dataset of all possible windows is so large that training rarely completed an entire epoch. For each bird, the validation data set was kept separate from the training and test data sets. For Bengalese finches and canaries we used: $(\texttt{val\_step} = 250, \texttt{patience} = 4, )$.

## Learning Curves

To estimate how much manually annotated training data is required to achieve a certain level of model performance, we generated learning curves that plot a metric such as frame error as a function of the amount of training data, as in the experiments shown in *Figures 3 and 4*. These experiments followed standard methods for benchmarking supervised machine learning algorithms, following good practices (*James et al., 2013*), such as training multiple replicates on separate subsets of the training data. Producing these learning curves where the dataset size is measured in duration required extra steps not needed for other tasks such as image classification. For each individual bird, we fit networks with training sets of increasing size (duration in seconds) and then measured performance on a separate, fixed test set. For each training replicate, audio files were drawn at random from a fixed-size total training set until the target size (e.g. 60 s) was reached. If the total duration of the randomly drawn audio files extended beyond the target, they were clipped at that target duration while ensuring that all syllable classes were still present in the training set. After training completed, we computed metrics such as frame error and syllable error rate on the held-out test set for each bird. As stated, we chose to use a totally separate fixed-size set, instead of e.g. using the remainder of the training data set, or generating multiple test sets in a $k$-fold validation scheme. We did this for two reasons: first, because computing metrics on relatively large test sets can be computationally expensive, and second, because we wanted to be sure that any variance in our measures across training replicates could be attributed to the randomly drawn training set, and not to changes in the test set.

In the case of Bengalese finches, we used training sets with durations {30, 45, 75, 120, 180, 480, 600}, training 10 replicates for each duration, with subsets drawn randomly from a total training set of 900 seconds for each individual bird. The duration of the fixed test set for each bird was 400 s. For canaries, we used training sets of durations {120, 180, 240, 300, 360, 420, 480, 540, 600, 660}, training seven replicates for each duration, with subsets drawn randomly from a total training set of 25,000 s for each bird. The duration of the fixed test set for each bird was 5000 s. For the point estimate of the model's asymptotic syllable error rate on canary song, we used a training set of 6000 s and a test set of 5000 s.

The method for generating learning curves as just described is built into the vak library and can be reproduced using the learncurve command in the terminal, along with the configuration files we shared.

## Comparison with a support vector machine model

In *Figure 3*, we compare performance of TweetyNet with a Support Vector Machine (SVM) model. We trained the model on a set of audio features first described in *Tachibana et al., 2014*. Feature extraction code was translated to Python from original Matlab code kindly shared by the author. Based on previous work (*Nicholson, 2016*), we used a Support Vector Machine with a radial basis function (RBF) kernel. To find good values for the kernel coefficient $\gamma$ and the regularization parameter $C$, we performed halving random search across a range of values ($\gamma$=(1e-9, 1e-3), $C$=(60,1e10), log uniform distribution). In initial tests, we found that values of $\gamma$ larger than 1.0 tended to produce pathological behavior where the model predicted one class for all features. We chose ranges for hyperparameter search that avoided this behavior. To carry out hyperparameter search we developed a pipeline in scikit-learn (*Pedregosa et al., 2011*; *Grisel et al., 2020*). In very rare cases, for two of the four birds, we needed to perform more than one run of the pipeline to find hyperparameters that did not cause it. To extract features and train models we adapted code from the hybrid-vocal-classifier library (*Nicholson, 2021b*), which provides a high-level interface to scikit-learn, and our pipeline including hyperparameter tuning was similarly built with scikit-learn code. All SVM models were trained on the exact same train-test splits used for training TweetyNet, by using dataset files generated by vak. This meant that for each training set duration there were 10 replicates trained for Bengalese finch song and seven replicates for canary song.

## Statistics

To compare syllable error rates, we used the Wilcoxon paired signed-rank test, a non-parametric alternative to the T-test. We computed the test once for each training set duration, using paired samples: same number of training replicates that were each trained on a randomly drawn subset of training

data, where the 'factor' within each pair of replicates was the model used, TweetyNet or the SVM. To test for homogeneity of variance, we used Levene's test.

## Additional analysis of model performance
### Percentage of errors near boundaries
In 'Simple post-processing greatly reduces syllable error rates' we estimate the percentage of errors near boundaries. By 'boundaries' we mean the onset and offset times of syllables when they are manually annotated. A distribution of syllable durations computed from these onsets and offsets shows that boundaries are not static. There are two sources of this variation: naturally occurring motor variability in birdsong, and an additional noise component added by audio segmentation and human annotators. This variance in turn gives rise to frame errors, where the ground truth annotation and a trained TweetyNet model disagree about which of the time bins should be assigned the 'background' label. These frame errors very close to boundaries are likely to have a much smaller impact on the syllable error rate than frame errors in the middle of syllables, because near the boundary they have no effect on the sequence of labels produced by segmenting the model output, and only a minor effect on the estimated onset and offset times. In the main text and in *Figure 5—figure supplement 1* we estimate the percent of all such frame errors occurring at these noisy syllable onset and offset boundaries. To do so, we computed for every onset and offset the number of frame errors within a fixed distance of two time bins that specifically involved disagreement between the ground truth annotation and the trained model on the the 'background' class.

### Errors in rare sequences
For all sequence of Bengalese finch syllables a-b we examined all possibilities for the following syllable and identified the most frequent sequence, a-b-**x**. Then, among all sequences a-b-**y** that are at least four times less frequent than a-b-**x**, we measured the frame error during the syllable **y**. This detailed analysis showed that there is a very small effect on rare variants. Namely, even if the sequence a-b-**y** appears 100–1000 times less frequently than a-b-**x** it does not incur high error rates in most cases. We use two statistical tests to quantify this claim. First, we measure the Pearson correlation between the relative frequency of the rare event (prob(a-b-**y**) divided by prob(a-b-**x**)) and the frame error in the syllable 'y' (the fraction of spectrogram time bins not labeled 'y' within that segment). Second, we divide the rare events to the more rare and more common (relative frequency smaller or larger than 1/8) and measure the fraction of rare events exceeding the median error rate. We use the binomial z-test to compare the fraction and show that the difference is not significant.

### Model output as syllable likelihoods
In *Figure 9*, we present model outputs one step prior to assigning the most likely label to each spectrogram time bin. At that stage, one before the *argmax(N)* step in *Figure 2*, the model output for a given time bin $t$ is a real-valued affinity $a(t,s) \in \mathcal{R}$ of all predefined syllable classes $s$. In *Figure 9* we convert these numbers to likelihoods by subtracting the minimum value and normalizing separately for each time bin $L(t,s) = \frac{a(t,s) - \min_{s'} a(t,s')}{\sum_{\sigma}[a(t,\sigma) - \min_{s'} a(t,s')]}$. This transformation was done for presentation only. Applying the commonly-used softmax transform ($x \rightarrow \frac{exp(x)}{\sum_{x} exp(x)}$) is equivalent since we only keep the maximal value.

## Analysis of behavioral data and predicted annotations
### Bengalese finch branch points
We analyzed the Bengalese finch song in *Nicholson et al., 2017* to determine whether we could replicate key findings about the stability of branch points from *Warren et al., 2012* as described in the main text, and, if so, whether we could recover that results from annotations predicted by TweetyNet.

To analyze statistics of branch points in the Bengalese finch song from *Nicholson et al., 2017*, we first identified candidate branch points by visual inspection of each birds' annotated song. Then, for each day of a bird's song, we counted all occurrences of transitions from one syllable class to another, that is bigrams. We placed these counts in a matrix where rows were the first syllable of the bigram ('from') and the columns where the second syllable ('to'), and then performed a row-wise normalization to produce a first-order Markov transition matrix, where the elements are transition probabilities

from one syllable class to another. We also applied a thresholding so that any elements in the matrix less than 0.002 were set to 0.

## Statistical test

To test whether transition probabilities were stable across days, we used a permutation test, replicating the analysis of *Warren et al., 2012*. We took all occurrences of a transition point across two days, and then for each permutation, swapped the label for which day it belong to, and then computed the transition probabilities for the permuted days. Using 1000 permutations, we generated a distribution of diffrences and then asked whether the observed difference was larger than this bootstrapped distribution.

## Analysis of predicted annotations

Before testing whether we could recover the result that branch points were stable from annotations predicted by TweetyNet, we measured the syllable error rate of the trained models that we would use to predict annotations. For 1 day of song from one bird (or60yw70), we realized that the manually-annotated set of songs was even smaller (200 s) than the test sets we used in the benchmarking section. We removed this day because it is not really an 'entire day' of song.

## Canary syntax model

### Shared template dependence on number of syllables in song

In each bird, we define an upper bound for repeating parts of songs using pairwise comparisons. For each song we examined all other songs with equal or larger number of syllables and found the largest shared string of consecutive syllables. The fraction of shared syllables is the ratio between the number of shared sequence and the number of syllables in the first, shorter, song. Then, we bin songs by syllable counts (bin size is 10 syllables) and calculate the mean and standard deviation across all pairwise comparisons. Results are shown in *Figure 1—figure supplement 2*.

### Probabilistic suffix trees

For each canary phrase type, we describe the dependency of the following transition on previous phrases with a probabilistic suffix tree. This method was described in a previous publication from our lab (*Markowitz et al., 2013*, *Markowitz, 2022a*). Briefly, the tree is a directed graph in which each phrase type is a root node representing the first order (Markov) transition probabilities to downstream phrases, including the end of song. The pie charts in *Figure 7*, *Figure 7—figure supplement 1*, and *Figure 7—figure supplement 2* show such probabilities. Upstream nodes represent higher order Markov chains that are added sequentially if they significantly add information about the transition.

### Model cross validation to determine minimal node frequency

To prevent overfitting, nodes in the probabilistic suffix trees are added only if they appear more often than a threshold frequency, $P_{min}$. To determine $P_{min}$ we replicate the procedure in *Markowitz et al., 2013* and carry a 10-fold model cross validation procedure. In this procedure the dataset is randomly divided into a training set, containing 90 percent of songs, and a test set, containing 10 percent of songs. A PST is created using the training set and used to calculate the negative log likelihood of the test set. This procedure is repeated 10 times for each value of $P_{min}$, the x-axis in *Figure 8a*. For data sets of different sizes (curves in *Figure 8a* x-axis in *Figure 8b*) the mean negative log-likelihood across the 10 cross validation subsets and across 10 data sets, y-axis in *Figure 8a*, is then used to find the optimal value of $P_{min}$ - the minimum negative log-likelihood that corresponds to the highest precision without over-fitting the training set. All PSTs in *Figure 7*, *Figure 7—figure supplement 1*, and *Figure 7—figure supplement 2* are created using the cross-validated $P_{min}$.

## Code Availability

The code implementing the TweetyNet architecture, and code to reproduce experiments and figures in this paper, are available here (version 0.7.1, 10.5281/zenodo.5823556).

To aid with reproducibility of our experiments, and to make TweetyNet more accessible to researchers studying birdsong and other animal vocalizations, we developed a software library, vak (*Nicholson and Cohen, 2021*), available here.

We also developed Python tools to work with the datasets and their annotation: (*Nicholson, 2021a*; *Nicholson, 2021c*; *Nicholson, 2021d*).

All software was implemented in Python, leveraging the following open-source scientific libraries, frameworks, and tools: attrs (*Schlawack, 2020*), dask (*Dask Development Team, 2016*), jupyter (*Kluyver et al., 2016*), matplotlib (*Hunter, 2007*; *Caswell et al., 2020*), numpy (*van der Walt et al., 2011*; *Harris et al., 2020*), pandas (*pandas development team, 2020*), scikit-learn (*Pedregosa et al., 2011*; *Grisel et al., 2020*), scipy (*Virtanen et al., 2020*), torch (*Paszke et al., 2017*), torchvision (*Marcel and Rodriguez, 2010*), seaborn (*Waskom et al., 2020*; *Waskom, 2021*), and tqdm (*da Costa-Luis, 2019*).

## Data Collection

### Use of existing datasets

Bengalese finch song is from two publicly-available repositories. Results in *Figures 3, 4 and 6* all make use of "Bengalese finch Song Repository" (*Nicholson et al., 2017*). For experiments in *Figure 4* we added song from four Bengalese finches in an additional dataset, (*Koumura, 2016*), and accompanied the paper (*Koumura and Okanoya, 2016*). Please see 'Annotation of Bengalese finch song' for a description of how we annotated that data. Supplementary figures with descriptive statistics of song also use datasets of Waterslager canary songs (*Markowitz et al., 2013*), Bengalese finch songs (*Koumura and Okanoya, 2016*) and Zebra finch songs (*Otchy et al., 2015*) generously shared by those authors.

### Domestic canary song screening

Birds were individually housed in soundproof boxes and recorded for 3–5 days (Audio-Technica AT831B Lavalier Condenser Microphone, M-Audio Octane amplifiers, HDSPe RayDAT sound card and VOS games' Boom Recorder software on a Mac Pro desktop computer). In-house software was used to detect and save only sound segments that contained vocalizations. These recordings were used to select subjects that are copious singers ($\geq 50$ songs per day) and produce at least 10 different syllable types.

### Domestic canary audio recording

All data used in this manuscript was acquired between late April and early May 2018 - a period during which canaries perform their mating season songs. Birds were individually housed in soundproof boxes and recorded for 7–10 days (Audio-Technica AT831B Lavalier Condenser Microphone, M-Audio M-track amplifiers, and VOS games' Boom Recorder software on a Mac Pro desktop computer). In-house software was used to detect and save only sound segments that contained vocalizations. Separate songs were defined by silence gaps exceeding 1 second.

## Acknowledgements

This study was supported by NIH grants R01NS104925, R24NS098536, and R01NS118424 (TJG) We thank J Markowitz and TM Otchy for sharing song datasets, and Nvidia Corporation for a technology grant (YC and Samuel J Sober lab). We also thank the Sober lab for providing compute resources and feedback on early versions of this work.

## Additional information

### Funding

| Funder | Grant reference number | Author |
|---|---|---|
| National Institute of Neurological Disorders and Stroke | R01NS104925 | Alexa Sanchioni<br>Emily K Mallaber<br>Viktoriya Skidanova<br>Timothy J Gardner |

| Funder | Grant reference number | Author |
|---|---|---|
| National Institute of Neurological Disorders and Stroke | R24NS098536 | Alexa Sanchioni<br>Emily K Mallaber<br>Viktoriya Skidanova<br>Timothy J Gardner |
| National Institute of Neurological Disorders and Stroke | R01NS118424 | Timothy J Gardner |

The funders had no role in study design, data collection and interpretation, or the decision to submit the work for publication.

## Author contributions

Yarden Cohen, Conceptualization, Data curation, Formal analysis, Investigation, Methodology, Resources, Software, Supervision, Visualization, Writing – original draft, Writing – review and editing; David Aaron Nicholson, Conceptualization, Data curation, Formal analysis, Investigation, Methodology, Resources, Software, Validation, Visualization, Writing – original draft, Writing – review and editing; Alexa Sanchioni, Emily K Mallaber, Viktoriya Skidanova, Data curation; Timothy J Gardner, Funding acquisition, Project administration, Resources, Writing – original draft, Writing – review and editing

## Author ORCIDs

Yarden Cohen ⓘ http://orcid.org/0000-0002-8149-6954
David Aaron Nicholson ⓘ http://orcid.org/0000-0002-4261-4719
Timothy J Gardner ⓘ http://orcid.org/0000-0002-1744-3970

## Ethics

All procedures were approved by the Institutional Animal Care and Use Committees of Boston University (protocol numbers 14-028 and 14-029). Song data were collected from adult male canaries (n = 5). Canaries were individually housed for the entire duration of the experiment and kept on a light-dark cycle matching the daylight cycle in Boston (42.3601 N). The birds were not used in any other experiments.

## Decision letter and Author response

Decision letter https://doi.org/10.7554/eLife.63853.sa1
Author response https://doi.org/10.7554/eLife.63853.sa2

---

# Additional files

## Supplementary files
• Transparent reporting form

## Data availability

Datasets of annotated Bengalese finch song are available here and here. Datasets of annotated canary song are available here. Model checkpoints, logs, and source data files are available here. Source data files for figure are in the repository associated with the paper here (version 0.7.1).

The following datasets were generated:

| Author(s) | Year | Dataset title | Dataset URL | Database and Identifier |
|---|---|---|---|---|
| Cohen Y | 2022 | Song recordings and annotation files of 3 canaries used to evaluate training of TweetyNet models for birdsong segmentation and annotation | https://doi.org/10.5061/dryad.xgxd254f4 | Dryad Digital Repository, 10.5061/dryad.xgxd254f4 |

*Continued on next page*

*Continued*

| Author(s) | Year | Dataset title | Dataset URL | Database and Identifier |
|---|---|---|---|---|
| Nicholson D, Cohen Y | 2022 | Model checkpoints, logs, and source data files | https://doi.org/10.5061/dryad.gtht76hk4 | Dryad Digital Repository, 10.5061/dryad.gtht76hk4 |
| Nicholson DA | 2022 | TweetyNet | https://doi.org/10.5281/zenodo.5823556 | Zenodo, 10.5281/zenodo.5823556 |

The following previously published datasets were used:

| Author(s) | Year | Dataset title | Dataset URL | Database and Identifier |
|---|---|---|---|---|
| Nicholson D, Queen JE, Sober JS | 2017 | Bengalese Finch song repository | https://figshare.com/articles/Bengalese_Finch_song_repository/4805749 | figshare, 10.6084/m9.figshare.4805749.v6 |

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
