## [Editor Report]

Animals create an enormous diversity of sounds. To study the neural basis or behavioral logic of animal communication, it is first necessary to categorize sounds into distinct types. Here, the authors create a novel neural network that includes an LSTM to enable automated annotation of massive birdsong datasets. This widely usable method will have a big impact in the birdsong field and, more generally, will provide an ascendant generation of scientists with yet another example of how machine learning methods are revolutionizing the rigorous study of animal behavior.

---

## [Decision Letter]

**Decision letter after peer review:**

Thank you for submitting your article "TweetyNet: A neural network that enables high-throughput, automated annotation of birdsong" for consideration by *eLife*. Your article has been reviewed by 3 peer reviewers, including Jesse H Goldberg as Reviewing Editor and Reviewer #1, and the evaluation has been overseen by Ronald Calabrese as the Senior Editor. The following individual involved in review of your submission has agreed to reveal their identity: Michael Brainard (Reviewer #3).

The reviewers have discussed the reviews with one another and the Reviewing Editor has drafted this decision to help you prepare a revised submission.

Summary:

Cohen and colleagues present an algorithm and software (TweetyNet) to facilitate automated propagation of labels to large sets of vocalizations, such as birdsongs. Currently, a variety of other imperfect methods including other automated algorithms as well as manual annotation are used to label the elements of songs. Especially for more complex songs such as those of the Bengalese finch and canary, as are the focus here, human labeling remains a gold standard, and this is problematic because of the amount of time required to label large data sets as well as the potential for inconsistencies in how humans apply labels. Hence, more advanced tools to facilitate the automated annotation of songs will be of significant value for the birdsong community and others working on analogous problems. The paper is very well written and the material that is presented in the main text is largely accessible to a non-technical audience of potential users. The authors provide useful statistics on the performance of Tweetynet in labeling syllables from both BFs and canary songs, and show that it does a better job than a previous approach that, like TN, relies both on local acoustic structure and sequencing of acoustic elements in applying labels. Overall, the authors do a compelling job of presenting TN as a useful tool that may facilitate new findings derived from analyzing larger datasets than is currently feasible. As the authors note, this approach is not perfect, but they provide a valuable framework and software package that will enable others to build on their approach.

Essential revisions:

(1) All reviewers agreed there was a lack of clarity over how TN is working "under the hood" and choice of parameters.

1.1. The architecture is clearly laid out in Figure 4 but a much better job of showing visualizations of how an example spectrogram is annotated will be helpful. For example, grayscale matrices (as in the bottom panels of Figure 8) could appear earlier in the paper with bengalese finch examples to really show readers how each ~2.7 ms timebin (frame) is being assessed, and how each frame's label is converted into a gap-annotated syllable-gap sequence. I'd like to see some kind of visualization that brings the reader from a syllable in a spectogram to the matrix, to the argmax function through to annotation will be helpful from the outset. The schematic could, if possible, clarify how the LSTM is operating over frames including how many it holds in memory.

1.2. Given what we know from ASR, the pooling step is likely very important for reducing unwanted input variance. Thus, more details are important here regarding the choice of the max operation, and a performance comparison to other potential types of pooling that justifies the choice. In particular, it is known that max pooling is particularly susceptible to over fitting (Goodfellow et al., 2013), and so some discussion of whether the likely gains that are provided by max pooling are worth the potential costs is warranted. Relatedly, the dimensions of the pooling operation are not clear. In speech recognition it is common to see pooling in frequency (but not time), whereas in image recognition pooling in both spatial dimensions is common. The exact form should be clarified, and (if possible) justified.

1.3 The use of the term 'majority vote' for the post-processing step was very confusing (for me at least) because at first read it appeared to be doing the same thing as the argmax function in the actual neural network. If I understand correctly, each 2.7 ms bin gets assigned a vector of length n (where n = number of syllables trained + silence). Syllables (a sequence of frames flanked by silence) will have a sequence of labels of length (sylldur)/2.7 ms. Argmax takes the most probable of those labels and annotates the syllable accordingly. So – intuitively, the argmax is basically taking a majority vote. In the postprocessing step, I think that after the network has annotated a given syllable, it will enforce that all frames inherit the label of the syllable. So if a 270 ms syllable was initially labeled 1111221111 (and called syllable 1) then the majority vote post-processing step will go back and override the frame labels as 1111111111. This probably helps with correct annotations at sequence transitions, but it's not exactly clear to me why this is the case unless the majority vote is occurring before a re-annotation step. If I understand this correctly (which I may not) – then the term majority vote is a bit misleading and should instead be called something like 'frame override.' If I am not understanding this correctly, then the authors should really clarify exactly what this postprocessing step is and why it works.

1.4. Relative influence of sequential versus local features. As I understand it, TN takes as input spectrograms in which each column of the spectrogram (corresponding to some small period of time) is a "bin", and a set of labels for each bin that are derived from how the corresponding segments of song were labeled by manual annotation. The network is then trained on "windows" That contain some large number (hundreds) of contiguous bins and their labels corresponding to many hundreds of milliseconds of song. I would appreciate some guidance regarding how performance depends on the relative influence of local acoustic structure (within a single bin, or within a single syllable) and more global sequential structure that depends on the specific sequence of syllables within-training windows. I assume that both of these will affect the labels that are applied to bins, but I have little intuition about their relative influences. Will this tend to work against the correct labeling of rare variants? For example if the sequence 'abc' is very prevalent in a song, and the sequence 'abb' is very rare, will the labeling of 'abb' be biased towards the labels 'abc'? More generally, it would be helpful to have a more pointed discussion of the degree to which TN performance depends on choices for parameters such as bin size and window size – is this something that a user might want to differently adjust for BF versus canary song versus other types of vocalizations?

1.5. Please describe the weight sharing strategy. Different frequency ranges are likely to have different local feature behaviors. For large numbers of hidden units it is likely that a full weight sharing strategy is most beneficial in the CNN, but this should be confirmed. If the authors investigated other weight sharing strategies, error rates for different weight sharing strategies should be compared under scenarios where the number of hidden units in the CCN are varied systematically. If not, they should provide a different justification for their current choice.

(2) Better clarify failure modes, and provide a more accurate description of TN limitations

All reviewers were surprised and skeptical at the suggestion that TN could be good for labeling juvenile songs, mouse vocalizations, and bat vocalizations. My impression is that the issues about human labeling are greatly exacerbated for these sorts of data sets, and that they have not proven to be very amenable to analyses that try to apply discrete categorical labels. So these sorts of data sets seem like they might be better approached with analysis schemes that do not rely on ascribing specific categorical labels. It would be helpful to elaborate on how TN would work in such cases, and perhaps illustrate with some juvenile finch songs or other highly variable and apparently non-categorical vocalizations (such as from mice or bats, see also point 3.1 below). Inclusion of such new analyses is not necessary for publication – but if such analyses are not conducted any claim of the utility of TN for these purposes should be dropped from the manuscript – to ensure that the results of the TN performance accurately match the claims made.

(3) Better integration with parallel work:

3.1. Beyond the forgoing justifications for the implementation of the specific networks presented in the paper, there is a broader question as to whether the supervised learning approach is an ideal solution for all or even most cases. A variety of unsupervised, self-supervised, and semi-supervised approaches are available. Of course, there are far too many to compare all of these, but some discussion of these alternatives at the end of the paper is warranted. In particular, DeepSqueak [Coffey et al., 2019] offers a similar CNN-driven front end but with unsupervised clustering of features from USVs (and see [Goffinet et al., biorxiv]). Other recent work [Sainburg et al., 2020] applies fully unsupervised approaches to find song syllables and directly compares automated clusters to hand labels for several songbird species (but not canaries). It seems these unsupervised approaches would be better suited for highly variable songs (e.g. babbling, budgerigar warble). If this is the case then the authors should explicitly say so – so readers know what conditions are suited for TN and what conditions are not.

3.2. The authors suggest that ref 39 represents state of prior art, but should elaborate: ref 39 shows substantially better performance than is attributed to it when given larger training sets or other variations in application, and my impression is that other methods may also do substantially better than ref 39, and therefore provide additional useful benchmarks for comparing TN performance. Some of these take an approach of segmenting syllables and then clustering and labelling syllables by their acoustic features, independently of sequential context. This forgoes the potential value of statistical regularities in sequence, but correspondingly avoids any potential issues associated with correct detection and labeling of syllables that occur in rare sequence variants, about which I have inquired above. I would be especially interested in seeing a comparison of TN relative to some version of those other approaches. For example, it would be nice to see how the Hybrid Vocal Classifier (HVC), developed by one of the co-authors (Nicholson), compares with TN in labeling of BF and canary songs. I would consider a comparison of performance between TN and HVC of value in assessing TN performance relative to other approaches that rely on individually segmented syllables. Another potential benchmark for comparison of TN performance would be the support vector machine approach in Tachibana, Oosugi, Okanoya (2014), which appears to suggest similar performance to TN, and may have a repository of labeled BF songs that would facilitate comparisons. It is not incumbent on the authors to test performance against all prior approaches, but without further evaluation against some alternatives, it would be appropriate to temper claims that TN is broadly better than other extant algorithms (at least in terms of syllable error rates for BF songs) and instead further empathize some of its specific differences and strengths relative to other methods (such as not requiring pre-segmentation, use of small training sets, implicit incorporation of sequential information, etc). In this respect the authors emphasize that TN was developed to deal with canary songs, and I would not be surprised if its performance relative to other algorithms was further differentiated when applied to these more complex (than BF) songs. In that respect, running some of the annotated canary songs through HVC or other labeling algorithms might further clarify the relative strengths of TN and could be a valuable addition.

---

## [Author Response]

Essential revisions:(1) All reviewers agreed there was a lack of clarity over how TN is working "under the hood" and choice of parameters.1.1. The architecture is clearly laid out in Figure 4 but a much better job of showing visualizations of how an example spectrogram is annotated will be helpful. For example, grayscale matrices (as in the bottom panels of Figure 8) could appear earlier in the paper with bengalese finch examples to really show readers how each ~2.7 ms timebin (frame) is being assessed, and how each frame's label is converted into a gap-annotated syllable-gap sequence. I'd like to see some kind of visualization that brings the reader from a syllable in a spectogram to the matrix, to the argmax function through to annotation will be helpful from the outset. The schematic could, if possible, clarify how the LSTM is operating over frames including how many it holds in memory.

Following reviewers' suggestions, we have revised what was Figure 4, now Figure 2. The figure now dedicates a separate panel (A) to illustrate the process of labeling an example spectrogram – including the processing of the deep network outputs (the grayscale matrices in what was Figure 8) to estimated syllable segments. In panel B the figure illustrates the properties of the constituent components (convolutional, recurrent, projection). We now make clear in the figure caption and in the methods that the bidirectional LSTM operates on the entire sequence in both directions.

1.2. Given what we know from ASR, the pooling step is likely very important for reducing unwanted input variance. Thus, more details are important here regarding the choice of the max operation, and a performance comparison to other potential types of pooling that justifies the choice. In particular, it is known that max pooling is particularly susceptible to over fitting (Goodfellow et al., 2013), and so some discussion of whether the likely gains that are provided by max pooling are worth the potential costs is warranted. Relatedly, the dimensions of the pooling operation are not clear. In speech recognition it is common to see pooling in frequency (but not time), whereas in image recognition pooling in both spatial dimensions is common. The exact form should be clarified, and (if possible) justified.

We thank the reviewers for pointing out this oversight we made when describing our model and how we developed it. We added details in the introduction and in the methods to address these concerns. Specifically we now cite relevant literature to justify our choice of max pooling, and underscore that previous work has shown no advantage of alternative types of pooling (Sainath et al., 2013). As the work we cite shows, all current state-of-the-art models for similar tasks use standard max pooling with no down-sampling in the time dimension. To make the dimensions of the pooling operation clear, we specifically state in the introduction that we use a stride of 1 in the temporal dimension to avoid down-sampling, and provide further details in the methods. To justify this choice, we cite appropriate literature which found that there was no advantage to using larger strides in the temporal dimension within pooling layers.

We describe the measures we took to avoid overfitting below. We agree with the reviewers that multiple methods to avoid overfitting exist. Specifically, Goodfellow 2013 discusses ‘Maxout’, an alternative to the ‘dropout’ technique. Both techniques are not related to max pooling and, to our knowledge, ‘Maxout’ is not widely adopted. Our understanding is that max pooling improves generalization (and thus helps avoid overfitting) because it encourages invariance to small shifts in features, but as the work we cited shows, there is little of this benefit seen when pooling in the time domain.

1.3 The use of the term 'majority vote' for the post-processing step was very confusing (for me at least) because at first read it appeared to be doing the same thing as the argmax function in the actual neural network. If I understand correctly, each 2.7 ms bin gets assigned a vector of length n (where n = number of syllables trained + silence). Syllables (a sequence of frames flanked by silence) will have a sequence of labels of length (sylldur)/2.7 ms. Argmax takes the most probable of those labels and annotates the syllable accordingly. So – intuitively, the argmax is basically taking a majority vote. In the postprocessing step, I think that after the network has annotated a given syllable, it will enforce that all frames inherit the label of the syllable. So if a 270 ms syllable was initially labeled 1111221111 (and called syllable 1) then the majority vote post-processing step will go back and override the frame labels as 1111111111. This probably helps with correct annotations at sequence transitions, but it's not exactly clear to me why this is the case unless the majority vote is occurring before a re-annotation step. If I understand this correctly (which I may not) – then the term majority vote is a bit misleading and should instead be called something like 'frame override.' If I am not understanding this correctly, then the authors should really clarify exactly what this postprocessing step is and why it works.

We agree with the reviewers that we did not clearly define and illustrate the term "majority vote". To clearly define this term and clarify its relationship to argmax, we made several changes. When addressing revision point 1.1, we added a graphic illustration of how this clean-up step is performed to the architecture diagram. In the figure we conceptually separated, as two panels, the deep neural network (DNN) whose parameters are learned from the training data, and the steps converting the DNN’s output to segmented syllables. The latter (panel A in the figure) graphically separates the step assigning a single label to each spectrogram time bin (argmax), and what we called ‘majority vote’ – the step taking continuous runs of non-silence time bins and assigning them one label by a majority vote among them.

We now dedicate a section (“Post-processing neural network output and converting it to annotations”) to similarly make clear the relationship between the argmax and the majority vote steps. In addition we adapted the reviewer's suggested language to explain how the majority vote overrides the label that the network predicts. The text reads: “Second, we then take a "majority vote" by counting how many times each label is assigned to any time bin in a segment, and then assigning the most frequently occurring label to all time bins in the segment, overriding any others.”

1.4. Relative influence of sequential versus local features. As I understand it, TN takes as input spectrograms in which each column of the spectrogram (corresponding to some small period of time) is a "bin", and a set of labels for each bin that are derived from how the corresponding segments of song were labeled by manual annotation. The network is then trained on "windows" That contain some large number (hundreds) of contiguous bins and their labels corresponding to many hundreds of milliseconds of song. I would appreciate some guidance regarding how performance depends on the relative influence of local acoustic structure (within a single bin, or within a single syllable) and more global sequential structure that depends on the specific sequence of syllables within-training windows. I assume that both of these will affect the labels that are applied to bins, but I have little intuition about their relative influences. Will this tend to work against the correct labeling of rare variants? For example if the sequence 'abc' is very prevalent in a song, and the sequence 'abb' is very rare, will the labeling of 'abb' be biased towards the labels 'abc'? More generally, it would be helpful to have a more pointed discussion of the degree to which TN performance depends on choices for parameters such as bin size and window size – is this something that a user might want to differently adjust for BF versus canary song versus other types of vocalizations?

We thank the reviewers for raising these important questions which we have addressed with extensive follow-up experiments to better understand the relative influence of local and global features. We did this in two ways: (1) by varying the size of the spectrogram window fed to the network and controlling the amount of global "context" that the network sees, and (2) by varying the size of the hidden state within the network’s recurrent layer and affecting the network’s ability to integrate information over time steps. We include the results in main figure 5. By running these experiments we show the network depends on both these variables, and that the hyperparameters we used for both Bengalese finches and canaries in other figures were not poorly chosen.

To address the question about errors in rare sequence variants we carefully identified and analyzed these variants in Bengalese finch data. For all sequences of syllables a-b we examined all possibilities for the following syllable and identified the most frequent sequence, a-b-x. Then, among all sequences a-b-y (y≠x) that are at least 4 times less frequent than a-b-x, we measured the frame error rate during the syllable y. This detailed analysis showed that there is a very small effect on rare variants. We include a description of this analysis in a methods section “Errors in rare sequences”. We include a supplementary figure to show this analysis. The figure shows that when using model parameters and training set durations, presented in the manuscript for well-trained models, there is no relation between error rates in syllables occurring in rare sequences and the rarity of the sequence. The figure also shows that in poorly-trained or parameter-impoverished models the likelihood of errors indeed increases in some rarely-occurring sequences. We believe this meets the reviewers’ prediction but since it only occurs in badly-formed or poorly-trained networks we do not include it in as a result but reference it from the main manuscript.

Finally, we have added language to the methods to further address questions about bin size and window size. We make it clear that users must choose a bin size that does not prevent the model from being able to segment the smallest occurring gaps in song. The results presented in figure 5 also help guide readers towards good starting points for choosing hyperparameters for Bengalese finches and canaries.

1.5. Please describe the weight sharing strategy. Different frequency ranges are likely to have different local feature behaviors. For large numbers of hidden units it is likely that a full weight sharing strategy is most beneficial in the CNN, but this should be confirmed. If the authors investigated other weight sharing strategies, error rates for different weight sharing strategies should be compared under scenarios where the number of hidden units in the CCN are varied systematically. If not, they should provide a different justification for their current choice.

We agree with the reviewers that it is important to make clear to researchers unfamiliar with neural networks that an advantage of convolutional neural networks is how they enable weight-sharing. We now describe how our architecture uses the common "full" weight sharing strategy. To our knowledge, studies of other forms of weight sharing did not show any clear improvements, and are not widely adopted; all state-of-the-art architectures for automatic speech recognition and audio event detection with convolutional layers simply use full weight sharing. In the methods, we cite "early" (Abdel-Hamid et al., 2014) work that specifically tested alternative weight sharing strategies for ASR, and note that this has not been widely adopted.

(2) Better clarify failure modes, and provide a more accurate description of TN limitationsAll reviewers were surprised and skeptical at the suggestion that TN could be good for labeling juvenile songs, mouse vocalizations, and bat vocalizations. My impression is that the issues about human labeling are greatly exacerbated for these sorts of data sets, and that they have not proven to be very amenable to analyses that try to apply discrete categorical labels. So these sorts of data sets seem like they might be better approached with analysis schemes that do not rely on ascribing specific categorical labels. It would be helpful to elaborate on how TN would work in such cases, and perhaps illustrate with some juvenile finch songs or other highly variable and apparently non-categorical vocalizations (such as from mice or bats, see also point 3.1 below). Inclusion of such new analyses is not necessary for publication – but if such analyses are not conducted any claim of the utility of TN for these purposes should be dropped from the manuscript – to ensure that the results of the TN performance accurately match the claims made.

We recognize that our writing was unclear here and that it is confusing for readers. We revised language about applying TN to other vocalizations, such as highly variable juvenile finch song, so that it does not sound as if we are suggesting that these vocalizations could somehow be cleanly categorized into discrete labels. Instead we make it clear that TN can potentially be used as a binary classifier that simply classifies each time bin as "vocalization" or "non-vocalization", in effect reproducing a user's cleaned-up segmentations, so that these segmented variable vocalizations can then be used for downstream analysis. To make it clear that these are not claims about our results, we have moved these suggestions to the "Ideas and Speculation" section of the discussion as suggested by *eLife* guidelines. We are in contact with other groups that are using TweetyNet for this exact purpose, but we feel that including such demonstrations is out of scope for the claims we make in the paper.

(3) Better integration with parallel work:3.1. Beyond the forgoing justifications for the implementation of the specific networks presented in the paper, there is a broader question as to whether the supervised learning approach is an ideal solution for all or even most cases. A variety of unsupervised, self-supervised, and semi-supervised approaches are available. Of course, there are far too many to compare all of these, but some discussion of these alternatives at the end of the paper is warranted. In particular, DeepSqueak [Coffey et al., 2019] offers a similar CNN-driven front end but with unsupervised clustering of features from USVs (and see [Goffinet et al., biorxiv]). Other recent work [Sainburg et al., 2020] applies fully unsupervised approaches to find song syllables and directly compares automated clusters to hand labels for several songbird species (but not canaries). It seems these unsupervised approaches would be better suited for highly variable songs (e.g. babbling, budgerigar warble). If this is the case then the authors should explicitly say so – so readers know what conditions are suited for TN and what conditions are not.

We agree with the reviewers that it is very important to compare and contrast our approach with others. We have revised the existing references to the papers that reviewers cited so that the discussion very clearly states the questions that the reviewers raised. We have added our current understanding of how these approaches are related and when they can work together.

3.2. The authors suggest that ref 39 represents state of prior art, but should elaborate: ref 39 shows substantially better performance than is attributed to it when given larger training sets or other variations in application, and my impression is that other methods may also do substantially better than ref 39, and therefore provide additional useful benchmarks for comparing TN performance. Some of these take an approach of segmenting syllables and then clustering and labelling syllables by their acoustic features, independently of sequential context. This forgoes the potential value of statistical regularities in sequence, but correspondingly avoids any potential issues associated with correct detection and labeling of syllables that occur in rare sequence variants, about which I have inquired above. I would be especially interested in seeing a comparison of TN relative to some version of those other approaches. For example, it would be nice to see how the Hybrid Vocal Classifier (HVC), developed by one of the co-authors (Nicholson), compares with TN in labeling of BF and canary songs. I would consider a comparison of performance between TN and HVC of value in assessing TN performance relative to other approaches that rely on individually segmented syllables. Another potential benchmark for comparison of TN performance would be the support vector machine approach in Tachibana, Oosugi, Okanoya (2014), which appears to suggest similar performance to TN, and may have a repository of labeled BF songs that would facilitate comparisons. It is not incumbent on the authors to test performance against all prior approaches, but without further evaluation against some alternatives, it would be appropriate to temper claims that TN is broadly better than other extant algorithms (at least in terms of syllable error rates for BF songs) and instead further empathize some of its specific differences and strengths relative to other methods (such as not requiring pre-segmentation, use of small training sets, implicit incorporation of sequential information, etc). In this respect the authors emphasize that TN was developed to deal with canary songs, and I would not be surprised if its performance relative to other algorithms was further differentiated when applied to these more complex (than BF) songs. In that respect, running some of the annotated canary songs through HVC or other labeling algorithms might further clarify the relative strengths of TN and could be a valuable addition.

We thank the reviewer for pointing out the need to better compare with other methods. We reached out to the authors of ref 39 and after discussion realized that there was a crucial difference between our methods and theirs. After communicating with the authors and doing further careful inspection of the annotation, we realized that the paper (ref 39) only selects certain sequences within much longer songs. As a result, the paper does not benchmark how well their algorithms perform when annotating entire songs. The central goal of our algorithm is to annotate entire songs end-to-end with a single model. Therefore we removed direct comparisons between our methods and theirs, although of course we still cite it as related work. Because we realized upon close inspection of the dataset that there were sequences that were left unannotated, we chose four birds and completely annotated all songs for those birds. We provide these complete annotations with the datasets we are submitting so that readers would still be able to use the song to benchmark ours and other models.

We carried the analysis the reviewers suggested. The first figure in the results now compares TweetyNet with a support vector machine (SVM) model built into the `hvc` library that uses features from Tachibana 2014. The reviewers are correct, it is clear that given cleanly segmented audio, SVM models can perform quite well for Bengalese finch song. Our results also make it quite clear why TweetyNet is needed for canary song, since there are no reliable segmentation algorithms, and even with cleanly segmented data by human annotators, the difference between TweetyNet and SVM performance is negligible. Accordingly, we have also tempered claims and changed language to be more specific about the strengths and weaknesses of different approaches, as suggested.